# Quantification of different flow components in a high-altitude glacierized catchment (Dudh Koshi, Himalaya): some cryospheric-related issues

Louise Mimeau[1], Michel Esteves[1], Isabella Zin[1], Hans-Werner Jacobi[1], Fanny Brun[1], Patrick Wagnon[1], Devesh Koirala[2], and Yves Arnaud[1]

[1]Université Grenoble Alpes, IRD, Grenoble INP, CNRS, IGE, Grenoble, France
[2]Nepal Academy of Science and Technology, NAST, Kathmandu, Nepal

*Correspondence to:* Michel Esteves (michel.esteves@ird.fr)

**Abstract.** In a context of climate change and water demand growth, understanding the origin of water flows in the Himalayas is a key issue for assessing the current and future water resources availability and planning the future uses of water in downstream regions. Two of the main issues in the hydrology of high-altitude glacierized catchment are (i) the limited representation of cryospheric processes controlling the evolution of ice and snow in distributed hydrological models and (ii) the difficulty to

define and quantify the hydrological contributions to the river outflow. This study estimates the relative contribution of rainfall, glacier and snow melt to the Khumbu River streamflow (Upper Dudh Koshi, Nepal, 146 km$^2$ , 43 % glacierized, elevation range from 4260 to 8848 m a.s.l.) as well as the seasonal, daily, and sub-daily variability during the period 2012-2015 by using the physically based glacio-hydrological model DHSVM-GDM (Distributed Hydrological Soil Vegetation Model - Glaciers Dynamics Model). The impact of different snow and glacier parameterizations was tested by modifying the snow albedo

parameterization, adding an avalanche module, adding a reduction factor for the melt of debris covered glaciers, and adding a conceptual englacial storage. The representation of snow, glacier, and hydrological processes was evaluated using three types of data (MODIS satellite images, glacier mass balances, and in situ discharge measurements). The relative flow components were estimated using two different definitions based on the water inputs and contributing areas. The simulated hydrological contributions differ not only depending on the used models and implemented processes, but also due to different definitions of

the estimated flow components. In the presented case study, ice and snow melt contribute each more than 40 % to the annual water inputs and 69 % of the annual stream flow originates from glacierized areas. The analysis of the seasonal contributions highlights that ice and snow melt as well as rain contribute to monsoon flows in similar proportions and that winter outflow is mainly controlled by the release from the englacial water storage. The choice of a given parametrization for snow and glacier processes, as well as their relative parameter values, has a significant impact on the simulated water balance: for instance, the

different tested parameterizations led to ice melt contributions ranging from 42 to 54 %. The sensitivity of the model to the glacier inventory was also tested demonstrating that the uncertainty related to the glacierized surface leads to an uncertainty of 20 % for the simulated ice melt component.

# 1  Introduction

The Himalayan mountain range is known for being the water tower of Central and South Asia (Immerzeel et al., 2010). Its high elevated glaciers and snow cover play an important role in the regional hydrological system (Kaser et al., 2010; Racoviteanu et al., 2013) and provide water resources for the population living in the surrounding countries (Viviroli et al., 2007; Singh et al., 2016; Pritchard, 2017).

In the Hindu Kush-Himalaya (HKH) region climate change is expected to cause shrinkage of the snow and ice cover (Bolch et al., 2012; Benn et al., 2012; Kraaijenbrink et al., 2017). Changes in glacier and snow cover runoff are likely to have a significant impact on the hydrological regime (Akhtar et al., 2008; Immerzeel et al., 2012; Lutz et al., 2014; Nepal, 2016). Development of tourism is also affecting the accessibility to water during the peak of tourist season. In the Everest region in Nepal water needs have increased within the past decades due to higher demand in water supply for tourists and hydroelectricity production, leading to water shortage during months with low flows (winter and spring) (McDowell et al., 2013). Understanding the past and present hydrological regime and more particularly estimating the seasonal contribution of ice melt, snow melt, and rainfall to outflows is thus a key issue for managing water resources within the next decades. Indeed, the quantification of the ice melt contribution enables to assess the proportion of water currently available which is coming from a long term accumulation in the glaciers, and thus to assess the annual decrease of the basin water storage due to glacier melt. Moreover, knowing the fraction of snow melt, ice melt, and rainfall to the river outflow, and understanding their hydrological pathways can give insights into how much water is currently seasonally delayed and how the seasonal outflow and the overall water balance might be impacted in the future when this delay changes or if the ratio snowfall to rainfall changes (Berghuijs et al., 2014).

Recent studies have estimated present glacier and snow melt contributions to the outflow in Nepalese Himalayan catchments (e.g., Andermann et al., 2012; Savéan et al., 2015; Ragettli et al., 2015) and simulated future hydrological regimes using glacio-hydrological models (Rees and Collins, 2006; Nepal, 2016; Soncini et al., 2016). Results have demonstrated large differences in the estimates of the contribution of glaciers to the annual outflows of the Dudh Koshi catchment in Nepal, which range from 4 to 60 % (Andermann et al., 2012; Racoviteanu et al., 2013; Nepal et al., 2014; Savéan et al., 2015).

One of the main sources of uncertainty in modelling the outflow of Himalayan catchments is the representation of cryospheric processes, which control the evolution of ice and snow-covered surfaces in hydrological models. For instance, the representation of the debris covered glaciers in glacio-hydrological models is a challenge. Debris covered glaciers represent about 23 % of all glaciers in the Himalaya-Karakoram region (Scheler et al, 2011). The debris layers have been expanding during the last decades due to the glacier recession (Shukla et al., 2009; Bhambri et al., 2011; Benn et al., 2012) and are expected to keep expanding in the near future (Rowan et al., 2015). Since the study of Østrem (1959) it is known that the debris thickness has a strong impact on the meltwater generation, which means that a good representation of the debris covered glaciers in glacio-hydrological models is essential for estimating the amount of meltwater generated in glacierized catchment in the Himalayas. Many other cryospheric processes such as the liquid water storage and transfer through glaciers, snow transport by avalanches or wind, glacial lake dynamics and snow albedo evolution are either very simplified or not at all represented by the models

(Chen et al., 2017). It is therefore important to estimate the impact of such simplified representations of cryospheric processes on modelling results.

Delineation of the glacierized areas is another key entry-element to the glacio-hydrological model. Glacier inventories are commonly used as forcing data to delineate glacierized areas in glacio-hydrological modelling studies. There are three global major glacier inventories such as the World Glacier Inventory (Cogley, 2009), GlobGlacier (Paul et al., 2009) and the Randolph Glacier Inventory (Pfeffer et al., 2014), and several regional glacier inventories in the HKH region (ICIMOD (Bajracharya et al., 2010), Racoviteanu et al. (2013)), showing substantial differences. These can be due to the definition of the glacierized area itself (Paul et al., 2013; Brun et al., 2017) as well as to the characteristics of the satellite image (date, resolution, spectral properties) used for the delineation (Kääb et al., 2015), and to difficulties related to the interpretation of satellite images for outlying the glaciers, especially when they are debris covered (Bhambri et al., 2011; Racoviteanu et al., 2013; Robson et al., 2015). Thus, the question whether the glacier delineation has a significant impact on the model results needs to be addressed.

These issues of the representation of cryospheric processes and of glaciers delineation in the hydrological modelling are addressed in the present study by (i) adapting the parameterization of the snow albedo evolution of DHSVM-GDM, in order to improve the simulation of the snow cover dynamics; (ii) implementing an avalanche module; (iii) introducing a melting factor for debris covered glaciers and (iv) testing the sensitivity of simulated outflows and flow components with respect to these modifications as well as to glacier delineation for three different outlines coming from different glacier inventories. Both in-situ measurements and satellite data were used for evaluating the outflow simulations as well as snow cover and glacier evolutions focusing on a small headwater catchment.

There are indeed several ways to define the glacier contribution to runoff (Radić and Hock, 2014): it can be either considered as the total outflow coming from glacierized areas, the outflow produced by the glacier itself (snow, firn and ice melt) or the outflow produced only by the ice melt. How the contributions to the outflow are defined adds further uncertainty to the estimation of the glacier contribution. The definition of the glacial contribution is dependent to the hydrological model (distributed or lumped, representation of glaciers and snow in the model) and cannot always be chosen. In the Dudh Koshi basin, Andermann et al. (2012); Racoviteanu et al. (2013); Savéan et al. (2015) estimated the fraction of the outflow produced by ice melt, whereas Nepal et al. (2014) defined the glacier contribution as the fraction of the outflow coming from glacierized areas. Here, flow components were estimated using two different definitions of the hydrological contributions for assessing their relative contributions to the total water balance. Finally, model results are analyzed at the annual, monthly, daily and sub-daily scale in order to explain the origin of the water flows and their seasonal and daily variations.

## 2 Study area

This study focuses on the Pheriche sub-catchment of the Dudh Koshi basin (outlet at coordinates 27.89° N, 86.82° E) located in Nepal on the southern slopes of Mt. Everest in the Sagarmatha National Park (SNP) (Fig. 1). The catchment area is 146 km$^2$ and its elevation extends from 4260 to 8848 m a.s.l.

Local climate is mainly controlled by the Indian summer monsoon (Bookhagen and Burbank, 2006) and is characterized by four different seasons: a cold dry winter from December to March with limited precipitation, a warm and moist summer with most of the annual precipitation occurring during the monsoon from June until September, and two transition seasons: the pre-monsoon season in April and May and the post-monsoon season in October and November (Shrestha et al., 2000). At 5000 m, the annual precipitation is around 600 mm and the mean monthly temperature ranges from -8.4°C in January to 3.5°C in July, according to temperature and precipitation data from the Pyramid EvK2 station (Fig. 2 and Table 1). The hydrological regime follows the precipitation cycle with high flows during the monsoon season, when most of the annual precipitation occurs, complemented by the melting of snow and ice, and low flows during winter.

Due to high elevation, vegetation in the catchment is scarce. The basin area is mainly covered by rocks and moraines (43 %) (Bajracharya et al., 2010) and glaciers (43 %) (Racoviteanu et al., 2013). Only 14 % of the basin area is covered by grasslands and shrublands. Glaciers belong to the summer-accumulation type (Wagnon et al., 2013) and are partially fed by avalanches (King et al., 2016; Sherpa et al., 2017). 60 % of the glaciers are located between 5000 and 6000 m a.s.l.. Debris-covered glaciers are found at low elevations mainly on the ablation tongues of the glaciers (Fig. 3). According to the Racoviteanu et al. (2013) glacier inventory, debris covered glaciers represent 30 % of the glacierized area with smaller melting rates at similar elevations than debris-free glaciers due to the insulating effect of the debris layer (Vincent et al., 2016).

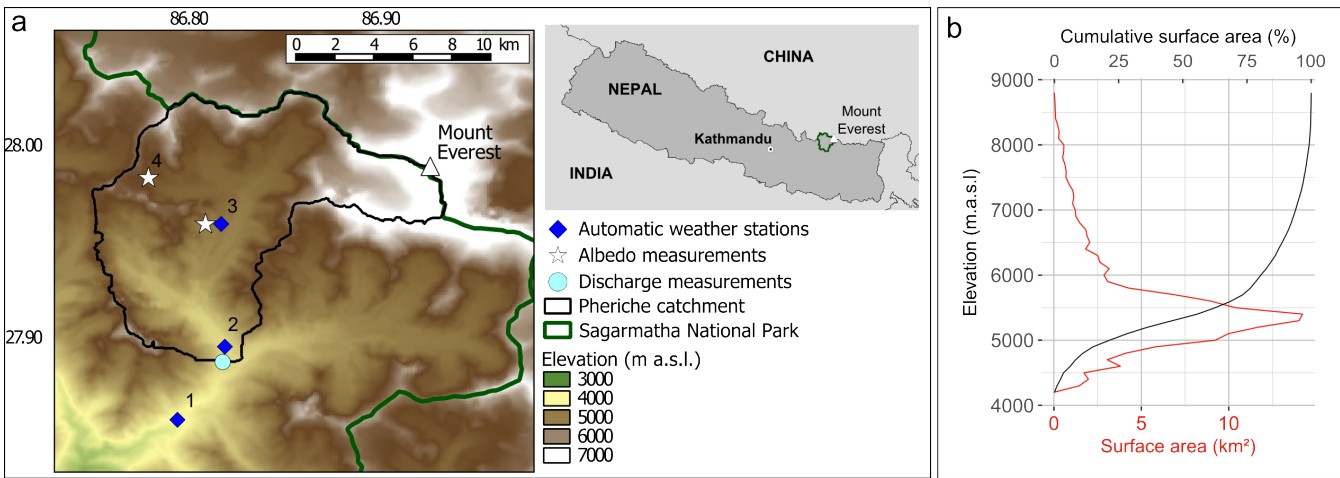

**Figure 1.** Study area : (a) Location map of Pheriche catchment (black) in the Sagarmatha National Park (green) in Nepal. Characteristics of the meteorological stations are summarized in Table 1. (b) Hypsometric curve of the Pheriche catchment.

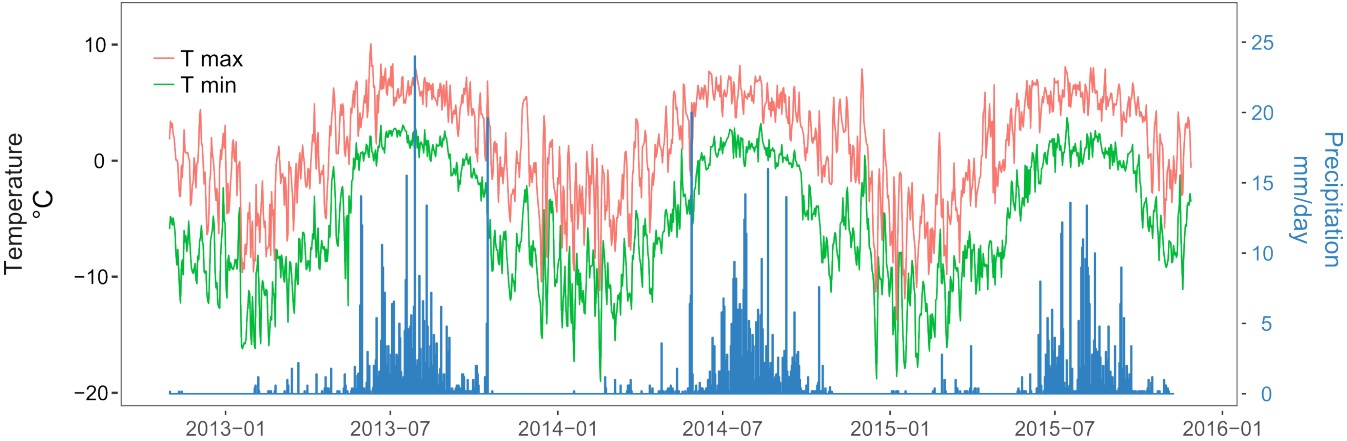

**Figure 2.** Daily minimal and maximal air temperature and daily precipitation measured at the Pyramid station.

## 3 Data and model setup

### 3.1 Database

To describe the topography of the study area, an ASTER DEM originally at 30 m resolution was resampled to a 100 m resolution. The SOTER Nepal soil classification (Dijkshoorn and Huting, 2009) and a landcover classification from ICIMOD (Bajracharya, 2014) were used for the soil and landcover description.

Meteorological data were available at hourly time steps from three automatic weather stations (AWS) located at Pangboche (3950 m a.s.l.), Pheriche (4260 m a.s.l.) and Pyramid (5035 m a.s.l.) (Table 1). Since December 2012, precipitation has been recorded at the Pheriche and Pyramid AWS by two Geonor T-200 sensors designed to measure both liquid and solid precipitation. Data were corrected for potential undercatch following the method used by Lejeune et al. (2007) and Sherpa et al. (2017). Precipitation at the Pangboche station was recorded with a tipping bucket. Air temperature, wind speed, relative humidity short-wave radiation and long-wave radiation measurements at Pheriche and Pyramid were provided by the EvK2-CNR stations.

Discharge measurements of the Khumbu River at Pheriche were obtained using a pressure water level sensor at a 30 minutes time step since October 2010.

| Glacier outline | Area | Satellite imagery used for delineation | Acquisition dates | Spatial resolution of the satellite images used for delineation |
|---|---|---|---|---|
| Racoviteanu et al. (2013) | Dudh Koshi, Langtang | ASTER, IKONOS-2 | 2003-2008 | 1 - 90 m |
| GAMDAM (Nuimura et al., 2015) | Asian glaciers | SRTM, LANDSAT | 1999-2003 | 30 - 120 m |
| ICIMOD (Bajracharya et al., 2010) | Nepal | IKONOS, LANDSAT, ASTER | 1992-2006 | 1 - 120 m |

**Table 2.** Glacier outlines characteristics

| N° | Name | Elevation (m) | Lat (°) | Lon (°) | Measured parameters | Manager |
|---|---|---|---|---|---|---|
| 1 | Pangboche | 3950 | 27.857 | 86.794 | T, P | IRD |
| 2 | Pheriche | 4260 | 27.895 | 86.819 | T, P, WS, RH, SWin | EvK2-CNR, IRD |
| 3 | Pyramid | 5035 | 27.959 | 86.813 | T, P, WS, RH, SWin, SWout, LWin | EvK2-CNR, IRD |
| 4 | Changri Nup | 5363 | 27.983 | 86.779 | SWin, SWout | GLACIOCLIM |

**Table 1.** Location of measurements. **T** air temperature, **P** precipitation, **WS** wind speed, **RH** relative humidity, **SWin** incoming shortwave radiation, **SWout** outgoing shortwave radiation, **LWin** incoming longwave radiation.

The MODImLab algorithm developed by Sirguey et al. (2009) was applied to MODIS reflectances data to obtain daily albedo and snow fraction satellite images for the period 2010-2015. We used the Sirguey et al. (2009) algorithm rather than the MOD10A1 500 m snow products because it generates daily regional snow cover images at 250 m resolution and applies corrections on atmospheric and topographic effects which makes the snow cover maps more realistic on mountainous areas. 27

cloud free Landsat8 images were used to generate snow maps at 30 m resolution between 1 November 2014 and 31 December 2015. A NDSI (Normalized-Difference Snow Index) threshold of 0.15 was taken to separate snow free and snow covered pixels on Landsat8 data as proposed by Zhu and Woodcock (2012). Daily snow cover maps were then retrieved from the MODImLab snow fraction product: areas with a snow fraction above 0.15 were defined as snow covered areas so that the MODImLab Snow cover area (SCA) matches the Landsat8 SCA on the common dates. For the rest of this study we call MODIS data albedo

and snow cover data obtained with the MODImLab algorithm. We also used snow albedo data from in-situ measurements at Pyramid and Changri Nup (Table 1).

For describing the glacierized area in the basin we compared three different glacier outlines available as vector layers for the Khumbu region: the glacier delineation proposed by Racoviteanu et al. (2013) specifically set up for the Dudh Koshi basin; the GAMDAM inventory covering the entire Himalayan range (Nuimura et al., 2015); and the ICIMOD inventory

(Bajracharya et al., 2010) (Fig. 3). The three outlines have been derived on different grids, from different datasets at different spatial resolutions and covering different temporal periods (see Table 2), thus leading at different results.

Mass balances estimated by Sherpa et al. (2017) for the clean-ice West Changri Nup and Pokalde glaciers located in the Pheriche basin (Fig. 3) were used as reference, as well as mean annual glacier mass balances calculated over the Pheriche basin area for the period 2000-2016 by Brun et al. (2017).

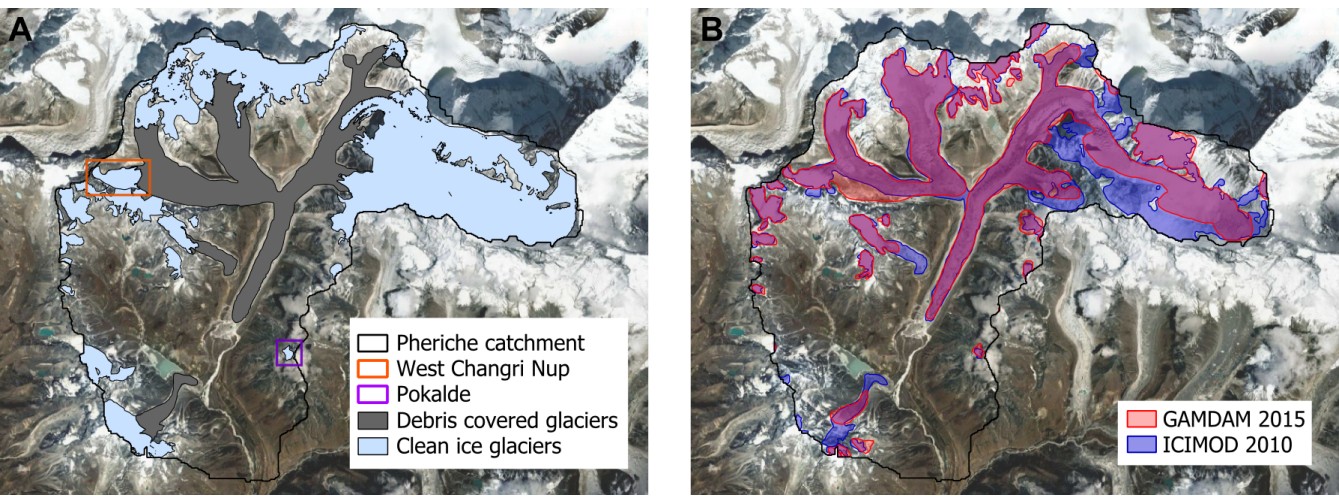

**Figure 3.** Glacier outlines in the Pheriche catchment. (a) Clean glaciers and debris-covered glaciers from Racoviteanu et al. (2013) and location of the clean ice West Changri Nup and Pokalde glaciers (b) GAMDAM (red) and ICIMOD (blue) glacier outlines

## 3.2 Glacio-hydrological modelling

### 3.2.1 General description of the model

The glacio-hydrological model DHSVM-GDM (Distributed Hydrological Soil Vegetation Model - Glaciers Dynamics Model) is a physically based, spatially distributed model which was developed for mountain basins with rain and snow hydrological regimes (Wigmosta et al., 1994; Nijssen et al., 1997; Beckers and Alila, 2004). A glacier dynamics module was recently implemented in DHSVM by Naz et al. (2014) to simulate glacier mass balance and the runoff production in catchments with glaciers, thus extending the application to ice dominated hydrological regimes. The resulting DHSVM-GDM simulates the spatial distribution and the temporal evolution of the principal water balance terms (soil moisture, evapotranspiration, sublimation, glacier mass balance, snow cover, and runoff) at hourly to daily time scales. It uses a two-layer energy and mass balance module for simulating snow cover evolution and a single layer energy and mass balance module for glaciers (Andreadis et al., 2009; Naz et al., 2014) and has been applied in a number of studies for snow and cold regions hydrology (e.g., Leung et al., 1996; Leung and Wigmosta, 1999; Westrick et al., 2002; Whitaker et al., 2003; Zhao et al., 2009; Bewley et al., 2010; Cristea et al., 2014; Frans et al., 2015). Distributed meteorological data (air temperature, precipitation, relative humidity, wind speed, and shortwave and longwave incoming radiation) are requested as input, as well as distributed geographical information (elevation, soil type, landcover, soil depth, and ice thickness).

### 3.2.2 Snow albedo parameterization

In the original DHSVM-GDM version, the snow albedo $\alpha_s$ [-] is set to its maximum value $\alpha_{smax}$ (to be fixed either by calibration or from observed albedo values) after a snowfall event and then decreases with time according to the following equations (Wigmosta et al., 1994):

$$
\begin{aligned}
\alpha_s &= \alpha_{smax}\,(\lambda_a)^{N^{0.58}} \quad if \quad T_s < 0 \\
\alpha_s &= \alpha_{smax}\,(\lambda_m)^{N^{0.46}} \quad if \quad T_s \geqslant 0
\end{aligned}
\tag{1}
$$

Where $N$ is the number of days since the last snowfall, $\lambda_a$ [-] and $\lambda_m$ [-] correspond to 0.92 and 0.70 for the accumulation season and the melt season, respectively, and $T_s$ is the snow surface temperature [°C].

MODIS albedo images and the albedo measurements from Pyramid and Changri Nup were used to analyse the decrease of snow albedo with age in various locations of our study area. Figure 4 compares the observed albedo decay as a function of time for snow events with at least three consecutive days without clouds after the snowfall with the albedo parameterization in DHSVM-GDM. Since the observed values are not well represented by the standard albedo decrease, the parameterization was replaced by Eq. 2, with a decay of the albedo when there is no new snowfall inspired by the ISBA model albedo parameterization (Douville et al., 1995) and with the fresh snow albedo modified as a function of the amount of snowfall:

$$
\begin{aligned}
\alpha_s &= (\alpha_{st-1} - \alpha_{smin})\exp(-cN) + \alpha_{smin} & if & \quad i_{snowfall} = 0 \text{ mm/h} \\
\alpha_s &= \max(0.6,\,\alpha_{st-1}) & if & \quad 0\text{ mm/h} < i_{snowfall} \leqslant 1\text{ mm/h} \\
\alpha_s &= \max(0.6,\,\alpha_{st-1}) + (\alpha_{smax} - max(0.6,\,\alpha_{st-1}))\,\tfrac{i_{snowfall}-1}{3-1} & if & \quad 1\text{ mm/h} < i_{snowfall} \leqslant 3\text{ mm/h} \\
\alpha_s &= \alpha_{smax} & if & \quad i_{snowfall} > 3\text{ mm/h}
\end{aligned}
\tag{2}
$$

Where $\alpha_{st-1}$ is the albedo from the previous time step, $\alpha_{smin}$ is the minimal snow albedo of 0.3 (estimated using the mean minimal albedo values observed at the station and on MODIS images), $N$ is the number of days since the last snowfall, $c$ is the coefficient of the exponential decrease [days$^{-1}$], and $i_{snowfall}$ the snowfall intensity [mm/h]. Since the observed decrease is dependent on elevation, the coefficient $c$ is calculated as a function of elevation according to Eq. 3:

$$
c = 20\exp(-0.001\,Z)
\tag{3}
$$

Where $Z$ is the elevation of the cell in m a.s.l.

The new function for the decrease of the snow albedo is also shown in Fig. 4.

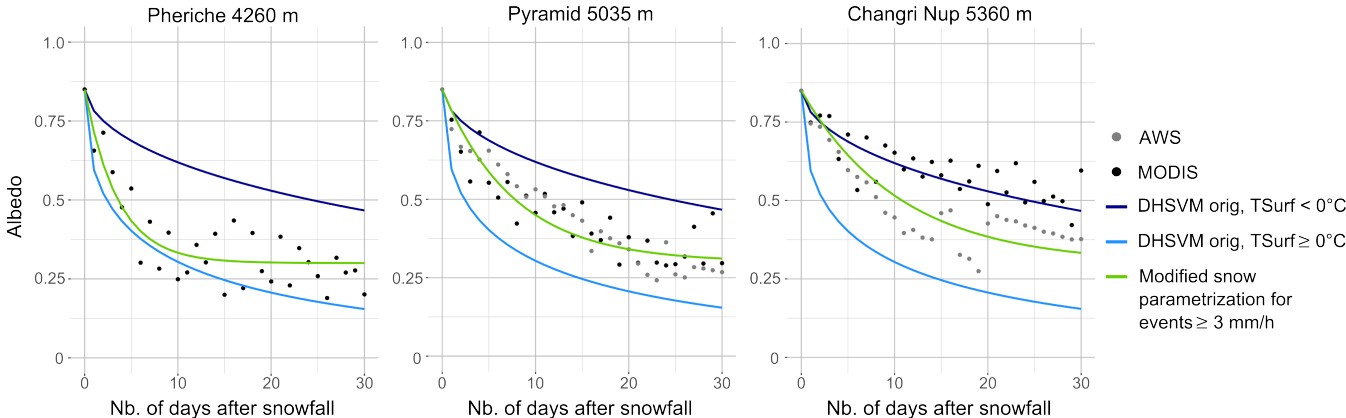

**Figure 4.** Original and modified parameterization of the snow albedo evolution in DHSVM-GDM and comparison with observed albedo data (2010-2015) in Pheriche, Pyramide and Changri Nup.

### 3.2.3 Avalanches parameterization

Transport of snow by avalanches is not represented in the original version of DHSVM-GDM. The absence of avalanches in the model can lead to an unrealistic accumulation of snow in steep high elevated cells, where the air temperature remains below 0 °C, and to a deficit of snow in the lower areas, where snow melt occurs during the melting season. The simulated water balance directly depends on the snow cover, thus not considering avalanches can lead to significant errors. In order to address these discrepancies, an avalanche module was implemented in DHSVM-GDM. The avalanche model transfers snow to downslope neighbour cells under the following conditions:

- if the terrain slope is steeper than 35 ° and the amount of dry snow water equivalent (total snow water equivalent minus liquid water content) is higher than 30 cm: 5 cm of snow water equivalent remains in the cell and the rest is removed by avalanches;

- if the terrain slope is less steep than 35 ° but the difference in snow water equivalent compared to the downslope neighbour cells is larger than 50 cm: 95 % of the difference is removed by avalanches.

The transfer of snow by avalanches is based on the surface runoff routing in DHSVM-GDM: at every time step starting from the highest cell of the DEM to the lowest, each cell can transfer snow to its closest downslope neighbour cells (between 1 and 4 cells). In case of several possible directions downstream, avalanche snow is distributed according to a ratio based on the slope of each of the directions. Within the same time step, the amount of snow in the receiving cells is actualised and the avalanches propagate downslope until the conditions cited above are no longer respected.

### 3.2.4 Glacier parameterization

Distributed ice thickness is derived from the terrain slope following the method described in Haeberli and Hölzle (1995).

In the original DHSVM-GDM version, glacier melt is instantaneously transferred to the soil surface, which is parameterized as bedrock under glaciers (Naz et al., 2014). This significantly underestimates the transfer time through glaciers. In this study, storage of liquid water inside glaciers was implemented by adding an englacial porous layer between the glacier and the bedrock allowing the liquid water storage within the glacier. Previous studies have shown the good performance of adding a

conceptual representation of the storage and drainage in the glaciers within glacio-hydrological models (e.g., Jansson et al., 2003; Hock and Jansson, 2006). The most widely adopted approach is based on a reservoir or a cascade of reservoirs with time-invariant parameters (e.g., Farinotti et al., 2012; Zhang et al., 2015; Hanzer et al., 2016; Gao et al., 2017). Here, storage of liquid water inside glaciers was implemented by adding an englacial porous layer between the glacier and the bedrock allowing the liquid water storage within the glacier. This englacial porous layer has a depth of 2 m and is characterized by a porosity of

0.8 and a hydraulic conductivity (vertical and lateral) of $3 \times 10^{-4}$ m/s (see Table A2). As in the previously cited studies, the parameters are kept constant through the simulations. They were optimized here according to the constraint of minimizing the differences (in terms of least squares) between the recession shape of the simulated hydrographs and the observed one

Moreover, since the standard DHSVM-GDM model does not take into account the impact of the debris layer on melting of the glaciers, the insulating effect of the debris layer is not represented. Here, we implemented a reduction factor for ice melt

generated in grid cells with debris-covered glaciers (see Sect. 3.3).

### 3.2.5 Quantification of the flow components

Quantifying the relative contributions of ice melt, snow melt, and rainfall in the river discharge at different time scales is a difficult task because hydrological models usually do not track the origin of water during transfer within the catchment (Weiler et al., 2018). There are also different ways of defining the origins of the streamflow. Weiler et al. (2018) lists three types

of contributions: 1) contributions from the source areas i.e. from each class of landcover, 2) contributions from the runoff generation (overland flow, subsurface flow, and groundwater flow), and 3) input contributions (ice melt, snow melt, and rain).

In this study, two different definitions were used to estimate the hydrological contributions. First, we estimate the contributions of ice melt, snow melt, and net rainfall to the total water input (definition 1) according to the following water balance equations (all the terms are fluxes expressed in [L/T]):

$$\mathrm{Input} = \mathrm{Icemelt} + \mathrm{Snowmelt} + \mathrm{RainNet} \tag{4}$$

$$\frac{dI_{\mathrm{wq}}}{dt} = \mathrm{GlAcc} - \mathrm{Icemelt} - \mathrm{SublIce} \tag{5}$$

$$\frac{dS_{\mathrm{wq}}}{dt} = P_{solid} - \mathrm{Snowmelt} - \mathrm{SublIce} - \mathrm{GlAcc} \tag{6}$$

$$\mathrm{RainNet} = P_{liquid} - E_{int} \tag{7}$$

$$\frac{d\mathrm{Storage}}{dt} = \mathrm{Input} - Q - E_T \tag{8}$$

Where $\frac{dI_{\mathrm{wq}}}{dt}$ and $\frac{dS_{\mathrm{wq}}}{dt}$ are the variations of the ice and snow storages, GlAcc is the amount of snow that is transferred to the ice layer by compaction on glaciers (Naz et al., 2014), SublIce and SublSnow are the amounts of sublimation from the ice and snow layers, $P_{solid}$ and $P_{liquid}$ are the amounts of solid and liquid precipitation, and $E_{int}$ is the amount of evaporation from

intercepted water stored in the canopy. It is worth noting that the sum of these contributions (Input) is not equal to the outflow at the catchment outlet $Q$ as it represents all liquid water reaching the soil surface (before infiltration and potential storage in the soils and glaciers($\frac{d\text{Storage}}{dt}$)) and before evapotranspiration ($E_T$)).

In order to evaluate the seasonal components of the outflow at the catchment's outlet, we also define the hydrological contributions as fractions of the outflow coming from the different contributing areas (definition 2):

– direct glacier contribution: direct runoff from glacierized areas,

– delayed glacier contribution: resurging melt water stored inside glaciers,

– direct snow contribution: direct outflow from snow covered non-glacierized areas,

– direct runoff: direct runoff from areas without snow and glaciers,

– subsurface and groundwater contribution: resurging water from the soil in non-glacierized areas resulting from infiltrated rainfall, snow melt, as well as upstream lateral subsurface flows.

These contributions are obtained from the amount of water reaching the soil surface simulated by DHSVM-GDM (see supplementary material). On each grid cell, this volume is a mixture of ice melt, snowmelt and rainfall and can either infiltrate into the soil or produce runoff. Definition 2 combines contributions from source areas (glacierized and non-glacierized areas) and contributions from runoff generation (direct runoff, englacial contribution, and soil contribution).

Figure 5 illustrates the two definitions of the different contributions to outflows. Definition 1 allows assessing the annual impact of glacier melt and snow melt on the water production, while Definition 2 describes the intra-annual routing of the water within the catchment. Moreover, using the two definitions allows to directly compare the results with other hydrological modelling studies in the Dudh Koshi basin, which have estimated glaciers contributions either from effective ice melt (Savéan et al., 2015; Ragettli et al., 2015; Soncini et al., 2016) or runoff from glacierized areas (Immerzeel et al., 2012; Nepal et al., 2014). Further, we assessed the impact of the definition of hydrological components on the estimated glaciers contribution.

Flow components were estimated for the period 2012-2015 at annual scale, on the basis of the glaciological year (from 1 December to 30 November), as well as monthly, daily, and sub-daily scales, in order to have a better understanding of the seasonal variation of the estimated hydrological contributions.

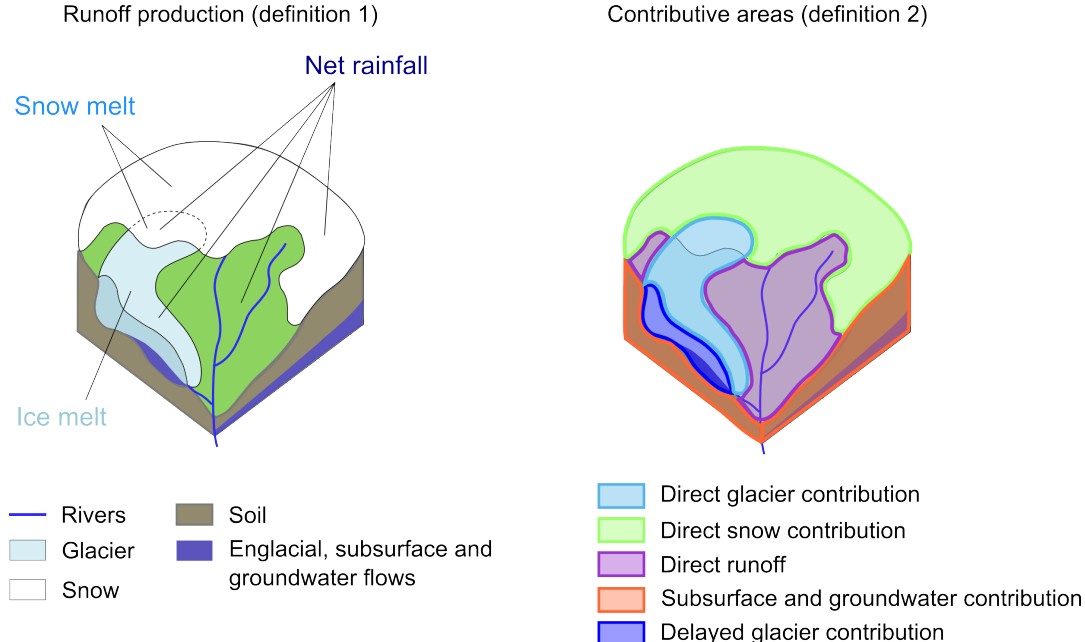

**Figure 5.** Definitions of the flow components.

### 3.3 Experimental set-up

Simulations were run with a 1 h time step and a spatial resolution of 100 m for the period from 1 November 2012 to 27 November 2015 corresponding to the period with most available meteorological and discharge data.

A soil depth map was derived from the DEM using the method proposed in the DHSVM-GDM documentation (Wigmosta et al., 1994). As a result, soil depth outside glacierized areas ranges between 0.5 and 1 m (glaciers are considered to lay on bedrock). All parameter values retained for the simulations (with no calibration) are summarized in Appendix A.

In order to test the impact of the representation of the cryospheric processes on the hydrological modelling, we performed simulations with the four following configurations:

- v0: original DHSVM-GDM snow and glacier parameterization;

- v1: modified snow albedo parameterization;

- v2: modified snow albedo parameterization and avalanche module;

- v3: modified snow albedo parameterization, avalanche module and melt coefficient for debris covered glaciers.

All four configurations were run with the Racoviteanu et al. (2013) glaciers outline. Concerning the melt of the debris-covered glaciers, we use a reduction factor of 0.4 as estimated by Vincent et al. (2016) from a study on uncovered and debris covered areas of the Changri Nup glacier.

Using configuration v3, we also tested the impact of using different glaciers outlines (the GAMDAM and ICIMOD inventories were also considered for simulations) and analyzed the sensitivity related to different values of parameters related to the cryospheric processes (see Table B1). Indeed, the debris-covered glacier melt reduction factor estimated in Konz et al. (2007), Nepal et al. (2014) and Shea et al. (2015) are respectively equal to 0.3, 0.33 and 0.47. Thus, values between 0.3 and 0.5 were also considered (in addition to the reference of 0.4). A sensitivity analysis of the englacial porous layer parameters (depth, porosity, and hydraulic conductivity) and avalanches parameters was also performed (see Table B1 for the tested parameter values) and the relative impact on the simulated hydrological response was discussed (see Sect. 5.3.2).

The sensitivity to the values of the englacial porous layer parameters (depth, porosity, and hydraulic conductivity), as well as to the values of the avalanches parameters, and to the soil depth is also analyzed in the discussion section of the manuscript

### 3.3.1 Model forcing

Meteorological data from the Pheriche and Pyramid stations (Table 1) were spatialized over the basin by an inverse distance interpolation method. Altitudinal lapse rates of precipitation and temperature were calculated at 1 h time step from data collected at Pangboche (3950 m a.s.l.), Pheriche (4260 m a.s.l.) and Pyramid (5035 m a.s.l.) (Fig. 6). Only significant lapse rates with $R^2$ values higher than 0.75 were retained for precipitation (43% of the dataset). For smaller $R^2$, the lapse rate is considered as not significant and thus set to 0.

In this study, the precipitation lapse rates show a large seasonal variability with daily lapse rates ranging from -41 to 9 mm km$^{-1}$. Precipitation decreases with elevation during the monsoon season and increases with elevation in winter: during the simulation period, we found 450 days (40 %) with no precipitation, 83 days (8 %) with a strictly negative lapse rate and 165 days (15 %) with a strictly positive lapse rate. Concerning temperatures, daily lapse rates range from -0.009 to +0.006 °C m$^{-1}$. We found only 10 days (1 %) showing a temperature inversion with a positive daily lapse rate.

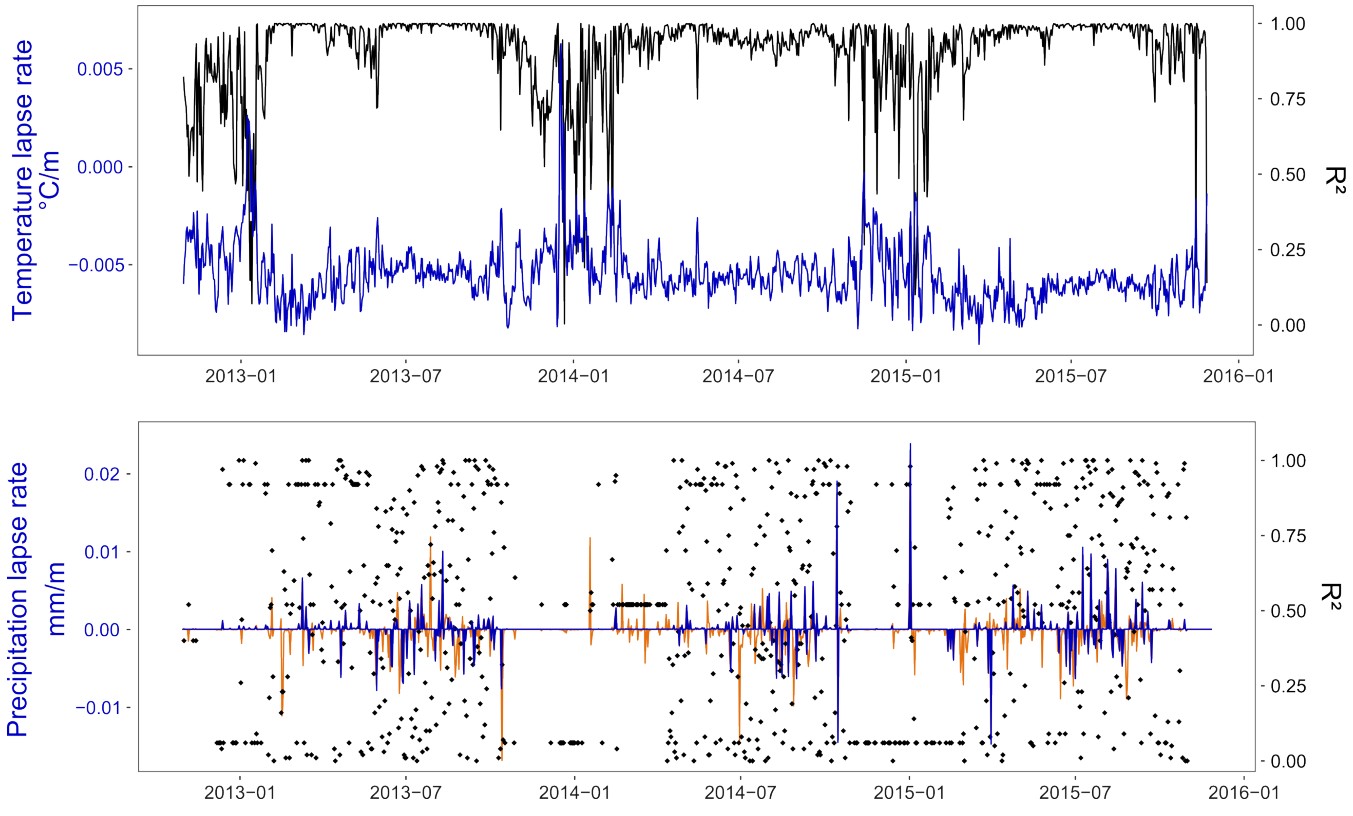

**Figure 6.** Daily temperature and precipitation lapse rates. Discarded precipitation lapse rates (with a $R^2$ <0.75) are represented in orange.

### 3.3.2 Model evaluation

A multi criteria evaluation was made considering simulated outflows, SCA and glacier mass balances. Discharge measurements at Pheriche station were used as reference for the evaluation of simulated outflows. A 15% confidence interval was retrieved as representative of the uncertainty of measured discharge. Nash-Sutcliffe Efficiency (NSE) (Nash and Sutcliffe, 1970) and

5    Kling-Gupta Efficiency (KGE) (Gupta et al., 2009) were chosen as objective functions and applied to daily discharges. The simulated SCA was evaluated in comparison to daily SCA derived from MODIS images. Because a large number of MODIS images suffer from cloud coverage, we only compared the simulated and observed SCA during days with less than 5 % of cloud cover on the catchment. The simulated glacier mass balances were evaluated at basin scale by a comparison with published regional geodetic mass balances and at local scale using available stake measurements on the clean ice West Changri Nup and

10    the Pokalde glaciers (Sherpa et al., 2017).

## 4 Results

### 4.1 Impact of the snow and glacier parameterizations on the simulated results

This section presents the simulation results obtained with the different configurations of the model DHSVM-GDM (configurations v0, v1, v2, and v3, see 3.2.5) and the analysis of the impact of the snow and glacier parameterizations on the simulated annual outflow, the daily SCA, annual glacier mass balances.

### 4.1.1 Annual outflow

Figure 7 represents the annual outflow and flow components (definition 1) simulated with the different model configurations, indicating the impact of each modification of the snow and glacier parameterization on the simulated annual outflow and flow components. Configuration v1 leads to a drastically increased outflow due to an enhanced ice melt component. Implementing the avalanche module (v2) reduces the ice melt component and increases the snow melt component by 21 %. Configuration v3 including debris-covered glaciers further reduces the ice melt, resulting in a simulated annual outflow close to the observations.

Figure 7 shows that configuration v2, which does not consider the debris-covered glaciers, overestimates the outflow at Pheriche with a mean bias of +32 % compared to the annual observed outflow.Without the debris layer, the ice melt component represents 817 mm, which is nearly twice the amount of ice melt obtained with v3 that includes debris-covered glacier melt.

The configuration with all three modifications (v3) gives results similar to the original parameterization of DHSVM-GDM (v0) in terms of glacier mass balance, improving slightly the annual outflow. The ice melt factor for debris covered glaciers and the avalanches compensate the increase of ice melt caused by the new snow albedo parameterization, but the modifications implemented in v3 impact the results for the flow components: on average, less ice melt and more snow melt are generated. Moreover, the configuration v3 modifies the seasonal variation of the outflow by increasing winter discharges and reducing monsoon discharges (not illustrated here) which improves the daily NSE and KGE (Table 3).

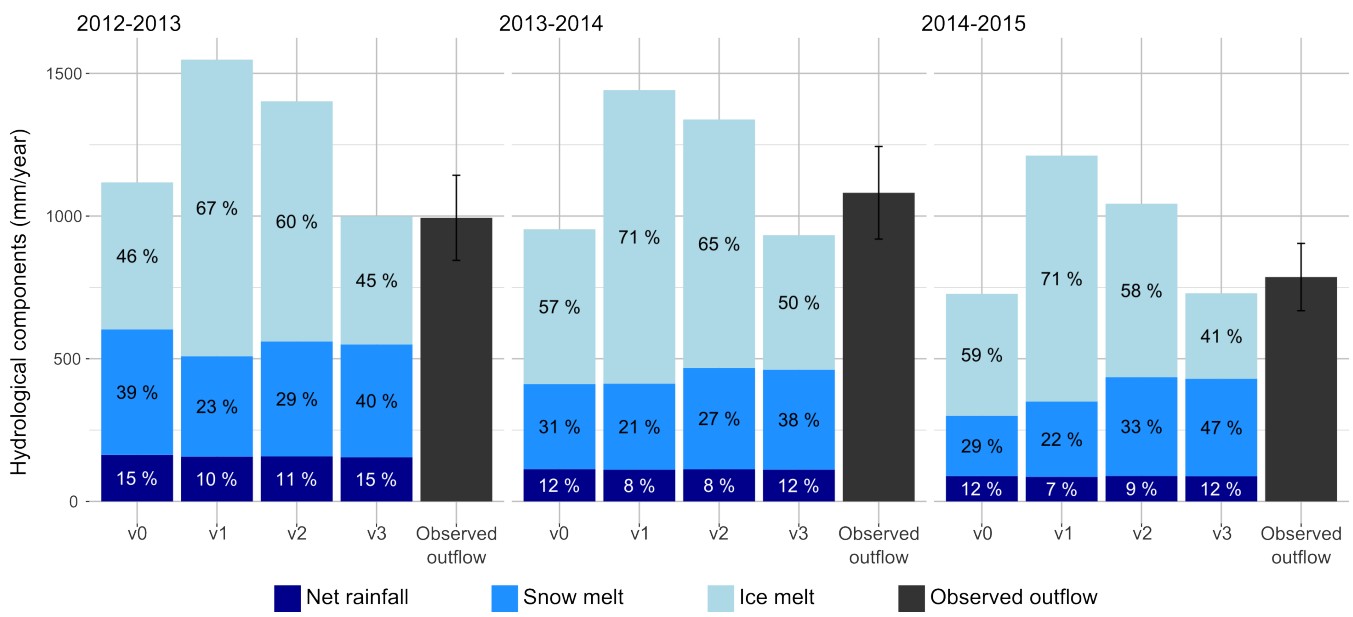

**Figure 7.** Simulated annual hydrological contributions (Definition 1) to Pheriche outflow for 3 glaciological years from 12/2013 to 11/2015

|      | v0   | v1   | v2   | v3   |
|------|------|------|------|------|
| NSE  | 0.87 | 0.53 | 0.74 | 0.91 |
| KGE  | 0.83 | 0.5  | 0.65 | 0.88 |

**Table 3.** NSE and KGE values calculated on the daily discharges on the period 2012-2015 for each model configuration.

### 4.1.2 Snow cover dynamics

Figures 8 and 9 compare the simulated snow cover area (SCA) and duration obtained with the configurations v0, v1, v2, and v3 to data derived from MODIS images. The SCA is strongly overestimated using the original parameterization v0: Figure 8 shows that after full coverage it does not decrease fast enough compared to MODIS data. Figure 9 demonstrates that the snow cover duration is over-estimated for the entire catchment area. This indicates that in the simulations snow does not melt fast enough with the original parameterization. Configuration v1 with the modified snow albedo parameterization (Eq. 2) accelerates the snow melt and improves the SCA simulation (Fig. 8). The RMSE between the simulated and observed SCA decreases from 29 % using v0 to 14 % using v1 and v2. Figure 9 shows that with configuration v1 in some areas located at high elevation the snow cover duration is underestimated. This bias is rectified in configuration v2 since the avalanche module transfers snow from high elevated and sloping cells downward and corrects the lack of snow observed with configuration v1 at the edges of the permanent snow cover (Fig. 9). The results for the SCA and snow cover duration using the configuration v3 show no difference compared to the configuration v2 since only the ice melt rate for debris covered glaciers is modified.

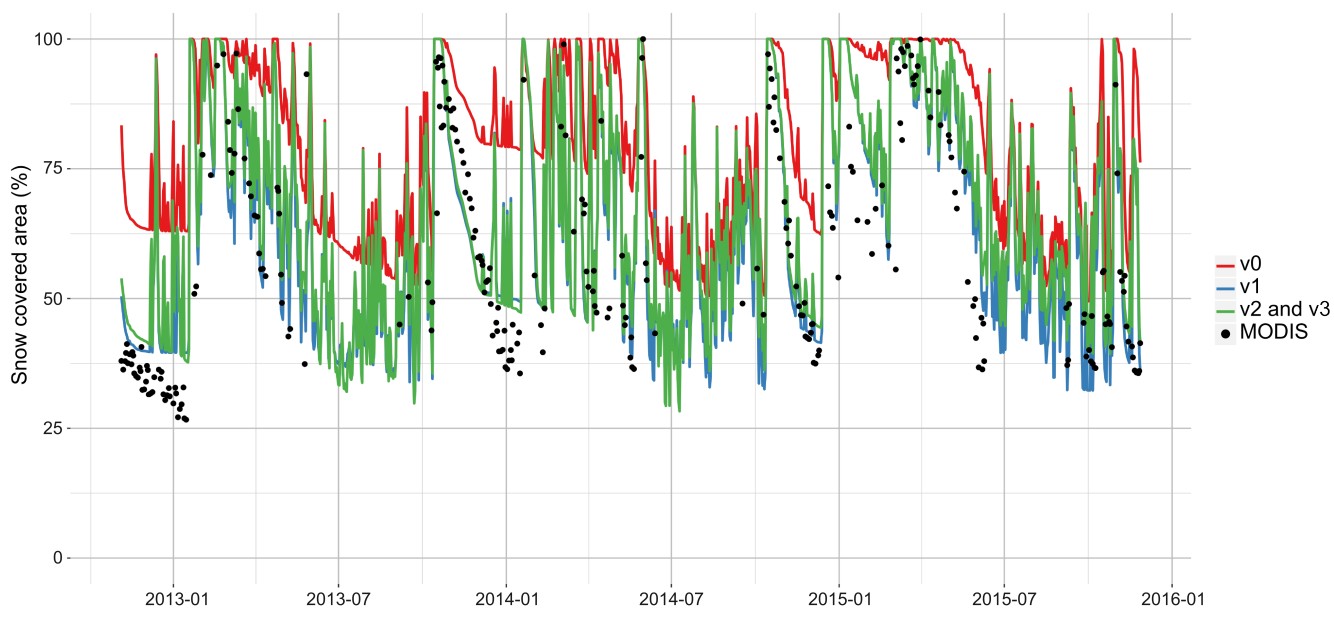

**Figure 8.** Comparison of the MODIS SCA and the simulated daily SCA with the four modelling configurations (v0, v1, v2 and v3) for the Pheriche catchment.

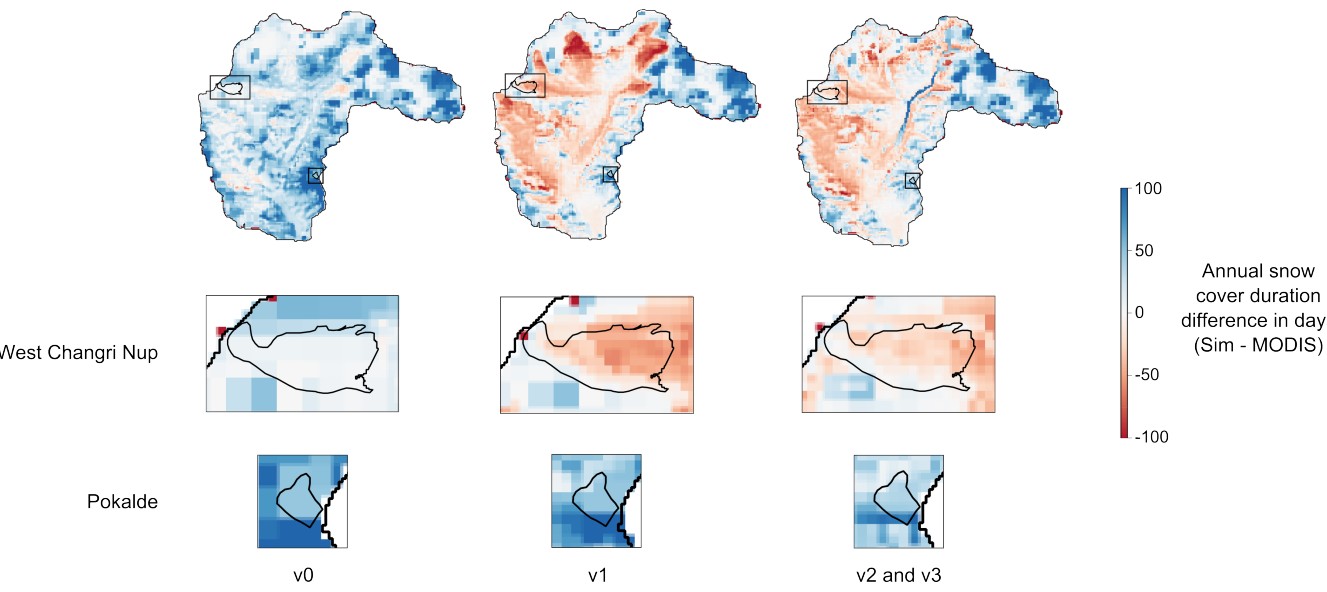

**Figure 9.** Difference between the mean annual snow cover duration simulated with DHSVM-GDM and derived from MODIS images (in days) for the Pheriche catchment (top panels), with a focus on West Changri Nup (medium panels) and Pokalde glaciers (bottom panels).

### 4.1.3 Glacier mass balances

Figure 10 compares the simulated mean annual glacier mass balances obtained with the different model configurations with mass balances determined with geodetic methods between 1999 and 2015 (Bolch et al., 2012; Gardelle et al., 2013; Nuimura et al., 2015; King et al., 2016; Brun et al., 2017). These geodetic mass balances range from -0.67 $\pm$ 0.45 m w.e.yr$^{-1}$ (2000-2008) (Nuimura et al., 2015) to -0.32 $\pm$ 0.09 m w.e.yr$^{-1}$ (2000-2015) (Brun et al., 2017).

Our results show that the snow parameterization has a significant impact on the simulated glacier mass balance. The mass balance simulated with v0 is on average -0.82 m w.e.yr$^{-1}$ and decreases to -2.02 m w.e.yr$^{-1}$ with the corrected snow albedo (v1) since the modified snow albedo parameterization accelerates the snow melt which leads to more uncovered ice and stronger glacier melt. The avalanche module (v2) adds snow on glaciers and increases the accumulation and, thus, reduces the glacier melt to -1.69 m w.e.yr$^{-1}$. Nevertheless the mass balance remains too negative compared to geodetic mass balances which suggests that the model produces too much ice melt. The implementation of debris-covered glaciers (v3) gives a mean annual glacier mass balance of -0.84 $\pm$0.14 m w.e.yr$^{-1}$ which is within the intervals of uncertainty and thus in good agreement with geodetic methods.

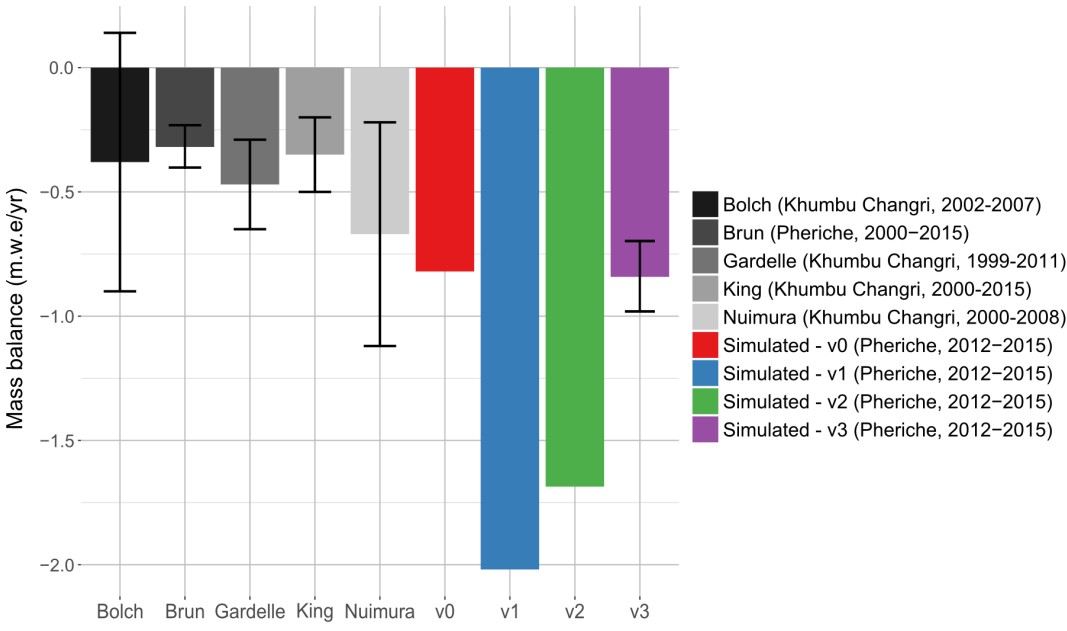

**Figure 10.** Mean annual glacier mass balances simulated with configurations v0, v1, v2 and v3. The error bar for configuration v3 represents the uncertainty related to the debris layer coefficient melt varying between 0.3 and 0.5.

We also evaluated the mass balance at the point scale. Figure 11 shows the simulated mass balances with parameteriza-
tions v0, v1, v2 and v3 versus the observed mass balances of the two debris-free glaciers West Changri Nup and Pokalde measured in-situ for the three glaciological years (2012-2015). Here, the configuration v3 gives the same results as the configu-

ration v2 because in configuration v3 only the ice melt rate on debris-covered glaciers is modified. The simulated mass balances vary according to the model configuration. With configuration v0, the model overestimates the point mass balances because of small snow melt rates (see also sectio 4.1.2).With configuration v1, the model overestimates the ice melt on the West Changri Nup glacier due to a lack of accumulation in the western part of the catchment and a too strong accumulation on the Pokalde

glacier (Fig. 9). The configuration v2 improves the simulated mass balance by transferring snow due to avalanches on the West Changri Nup glacier and by removing exceeding snow accumulation on the Pokalde glacier. For the Pokalde glacier, the mass balances simulated with configuration v3 show a larger variability than the mass balances simulated with configuration v0, but the point mass balances are spread around the diagonal axis which leads to a bias ten times smaller (mean bias of 1 m with v0 and 0.1 m with v3).

The results at basin scale and point scale show that the snow parameterization has a strong impact on the simulated glacier mass balance and that the new snow albedo parameterization and the avalanching module clearly improve the simulated glacier mass balance on debris-free glaciers. Nevertheless, regarding point mass balances, the agreement is far from being perfect, due either to simulation errors (including errors depending on the interpolated input fields and errors induced by the representation of slopes and expositions by the DEM) and/or from a lack of representativeness of the measurements.

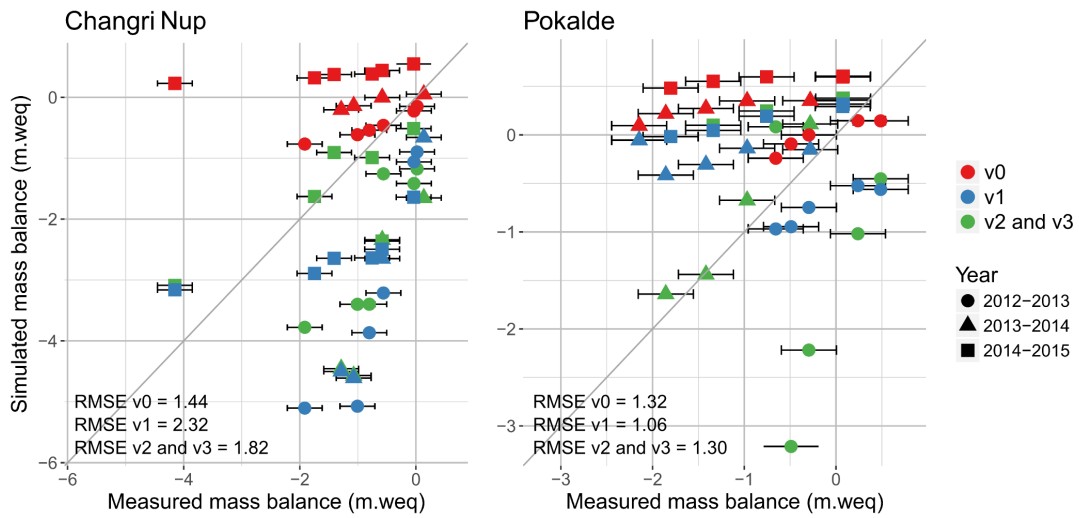

**Figure 11.** Annual simulated and measured point mass balances on West Changri Nup (left panel) and Pokalde (right panel) glaciers; also shown is the 1:1 line.

**4.2   Simulated outflow and flow components**

This section presents the outflows and flow components simulated in the Pheriche basin during the period 2012-2015 with the modified version of DHSVM-GDM (configuration v3). The simulation results are analysed using two different definitions of the flow components (definitions 1 and 2, see 3.2.5).

#### 4.2.1  Annual simulated outflow and hydrological contributions

The annual outflows simulated with the new parametrization of the model (configuration v3) are in good agreement with the annual observed outflows since they remain within the 15 % interval of estimated error (Fig. 12 and Table 4).

The results show an inter-annual variability of the flow components. During the period 2013-2015, the ice melt component ranged from 41 to 50 %, the snow melt component from 37 to 47 % and the net rainfall component from 12 to 16 %. These variations are related to the meteorological annual variability. The amount of rainfall decreased from 2013 to 2015 and explains the decrease of the net rainfall components from 155 mm in 2013 to 88 mm in 2015. The snow melt component is higher in 2013 because of warmer pre-monsoon and monsoon seasons. The ice melt component is mainly controlled by the amount of winter snowfall. In 2014 a low amount of snowfall was observed, so the snowpack melted more rapidly and the glaciers started melting earlier. In contrast, 2015 was a year with a lot of winter snowfall, which delayed the beginning of the glacier melt and explains the lower ice melt component. The losses by evaporation and sublimation are rather constant through the simulation period ranging from 140 to 150 mm/yr.

The runoff coefficients (ratio between the annual outflow and annual precipitation) were on average equal to 1.4, which means that a considerable amount of water is withdrawn each year from the catchment through ice melt (eventually in the form of a delayed groundwater flow).

On average, we find that the outflow is mainly produced by meltwater as 46 % of the annual water input is due to ice melt and 41 % to snow melt (definition 1). The contributions estimated according to definition 2 show the importance of infiltration and subsurface flows in the water balance since more than 40 % of the outflow was coming from water infiltrated in glaciers and more than 20 % from subsurface and groundwater flows generated outside the glacier covered area.

The choice of the definition of the hydrological components leads to different perceptions of the glacier contribution to the outflow. The glacier contribution to the total outflow is 69 % if the contribution from the entire glacierized area (i.e. contributions of ice melt, snow melt and net rainfall) is considered like in definition 2. However, the contribution from ice melt alone, included in definition 1, corresponds to only 46 % of the water input.

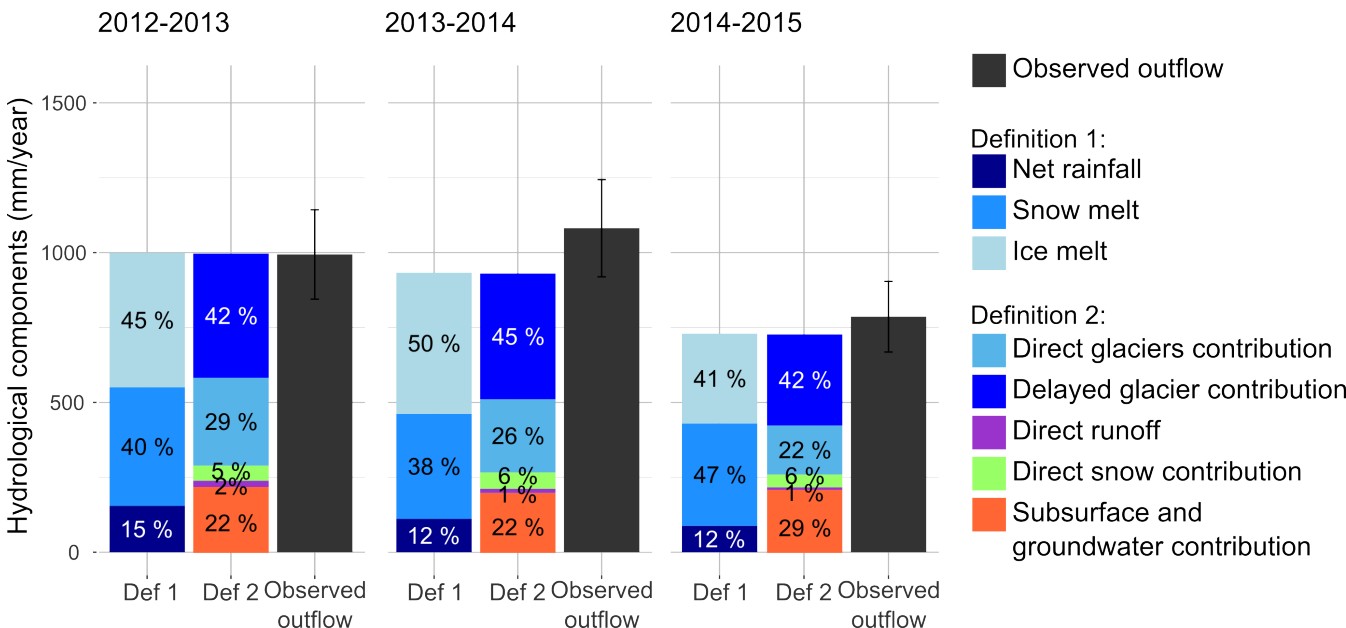

**Figure 12.** Simulated annual hydrological contributions to Pheriche outflow for the two definitions of the flow components (Definition 1 and Definition 2) and for 3 glaciological years from 12/2013 to 11/2015.

|                                  | 2013        | 2014        | 2015       |
|----------------------------------|-------------|-------------|------------|
| **Total precipitation (mm)**     | **708**     | **644**     | **683**    |
| Snowfall (mm)                    | 501         | 492         | 561        |
| **Qobs (mm)**                    | **994**     | **1081**    | **786**    |
| Qobs ±15% (mm)                   | 845 - 1143  | 919 - 1244  | 668 - 904  |
| **Qsim (mm)**                    | **999**     | **933**     | **729**    |
| Bias (%)                         | +1          | -14         | -7         |
| Evapotranspiration (mm)          | 61          | 48          | 43         |
| Sublimation (mm)                 | 91          | 96          | 97         |
| **Flow components (Definition 1)** |           |             |            |
| Net rainfall (mm)                | 164         | 117         | 93         |
| Snow melt (mm)                   | 420         | 368         | 362        |
| Ice melt (mm)                    | 476         | 496         | 317        |
| **Flow components (Definition 2)** |           |             |            |
| Direct glacier contribution (mm) | 293         | 244         | 164        |
| Delayed glacier contribution (mm)| 414         | 420         | 304        |
| Direct runoff (mm)               | 21          | 14          | 9          |
| Direct snow contribution (mm)    | 51          | 55          | 43         |
| Subsurface and groundwater contribution (mm) | 220 | 200      | 211        |

**Table 4.** Annual hydrological balance simulated with configuration v3 for 3 glaciological years from December 2013 to November 2015.

#### 4.2.2 Seasonal variations of the flow components

Figure 13 presents the daily simulated discharges simulated with configuration v3 and the flow components estimated with the two different definitions. Daily discharges were well simulated for 2012-2013 and 2014-2015 by the model, with NSE equal to 0.91 and KGE equal to 0.88. However, the outflow is under-estimated by the model during the monsoon season in 2014.

5    The simulated total water input (i.e. the sum of snow melt, ice melt and net rainfall) is always higher than the simulated outflow at the catchment outlet before the monsoon season (from February to June) and lower during post-monsoon and winter seasons. This is mainly due to glacier melt water stored inside the glaciers during pre-monsoon and monsoon seasons and continuing surging during winter, as well as to changes in the soil water storage (Fig. 13b and 14b).

Figure 14 shows the mean monthly flow components averaged over the simulation period. From February to May-June,

10   the water input is entirely controlled by snow and ice melt (snow melt between 50 and 60 % ice melt between 40 and 48 %) Fig. 14a). The net rainfall, snow melt and ice melt absolute contributions are at their maxima in July and August during the monsoon season. During these two months, 24 % of the runoff is generated by net rainfall, 37 % by snow melt, and 38 % by ice

melt. From October to January, the runoff is produced by ice melt (up to 80 % in December) and snow melt (between 20 and 30 %). Groundwater and englacial water represent a significant fraction of the monthly outflow as they contribute more than 50 % of the outflow during the monsoon season and can contribute up to 90 % during winter Fig. 14b). Direct contributions from glacierized areas, snow areas, and direct runoff are highest during the monsoon season, when the englacial and soil storage

5   is saturated.

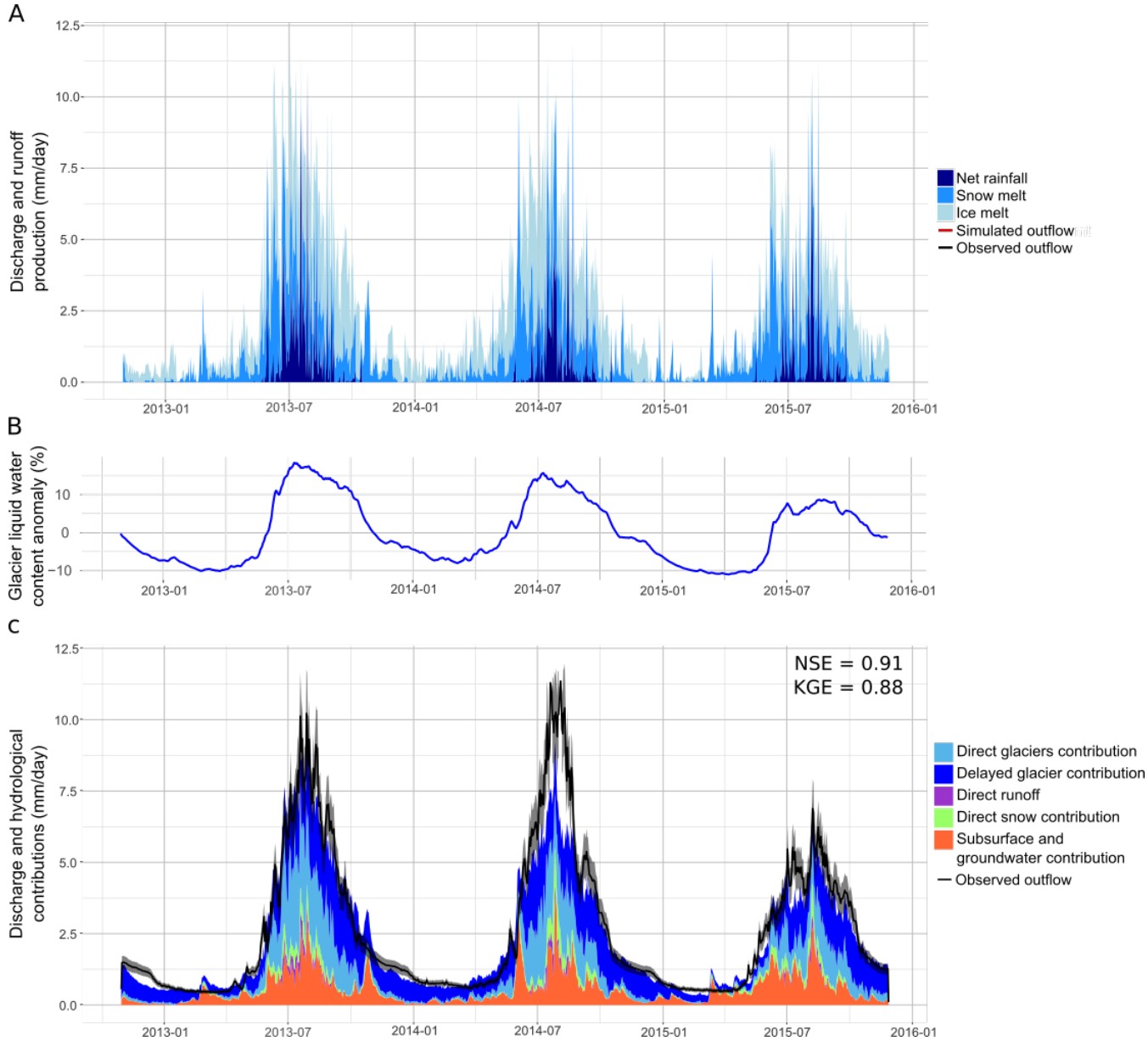

**Figure 13.** Daily discharges and flow components simulated with configuration v3: (a) production of ice melt, snow melt and net rainfall (note that the sum of the flow components represent the total water input and is not equal to the discharge at the catchment outlet, see definition 1, Sect. 3.2.5), (b) relative difference between the simulated glacier liquid water content and the inter-annual mean (c) hydrological contributions to the outflow (definition 2, Sect. 3.2.5). Observed discharges are represented by the black line with a 15 % interval of error.

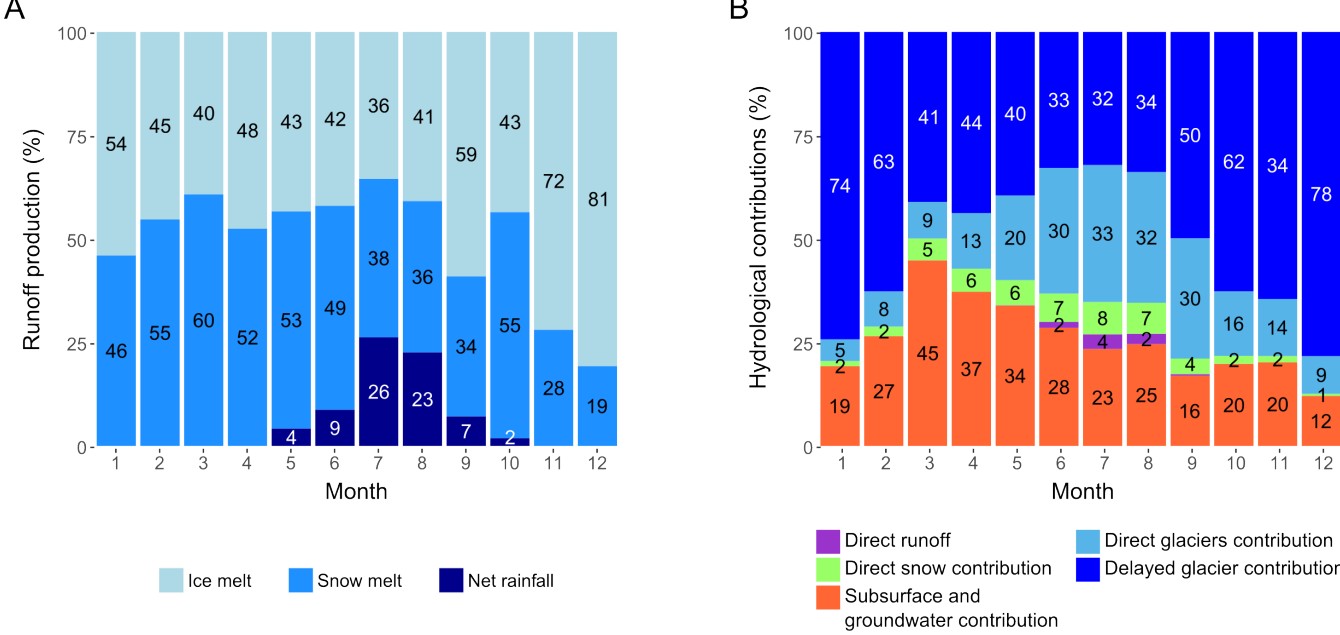

**Figure 14.** Average monthly contributions to the water input (definition 1, Sect 3.2.5) (a) and hydrological contributions (definition 2, Sect. 3.2.5) (b) simulated with configuration v3 for the years 2012-2015.

### 4.2.3 Diurnal cycle

Figure 15 presents the diurnal cycles of precipitation and hydrological components averaged for each considered season (winter, pre-monsoon, monsoon, and post-monsoon) obtained with configuration v3. During winter, pre-monsoon, and post-monsoon, the observed outflow is rather constant during the day, with a weak peak around noon when the temperature is at its maximum. During this period, almost all of the precipitation is in the form of snowfall leading to no direct response for the outflow. The peak around noon can be explained by snow melt or the melting of small frozen streams. During the monsoon season, there is a strong diurnal cycle of the precipitation with a maximum occurring during late afternoon or at night causing a peak in the discharge around midnight.

The model simulates ice and snow melt during day time with a maximum at noon as expected. Except for the monsoon season, it seems to simulate accurately the baseflow during night without melt production: the discharge is rather controlled by the release of the glacier and soil storage. However, the model simulates a peak of discharge around 14 h originating mostly from glacierized areas, two hours after the maximum of ice and snow melt, which does not correspond to observed discharges. At daily and longer time scales the water balance is correctly simulated. However, at a sub-daily scale the model responds too quickly to the snow and ice melt production.

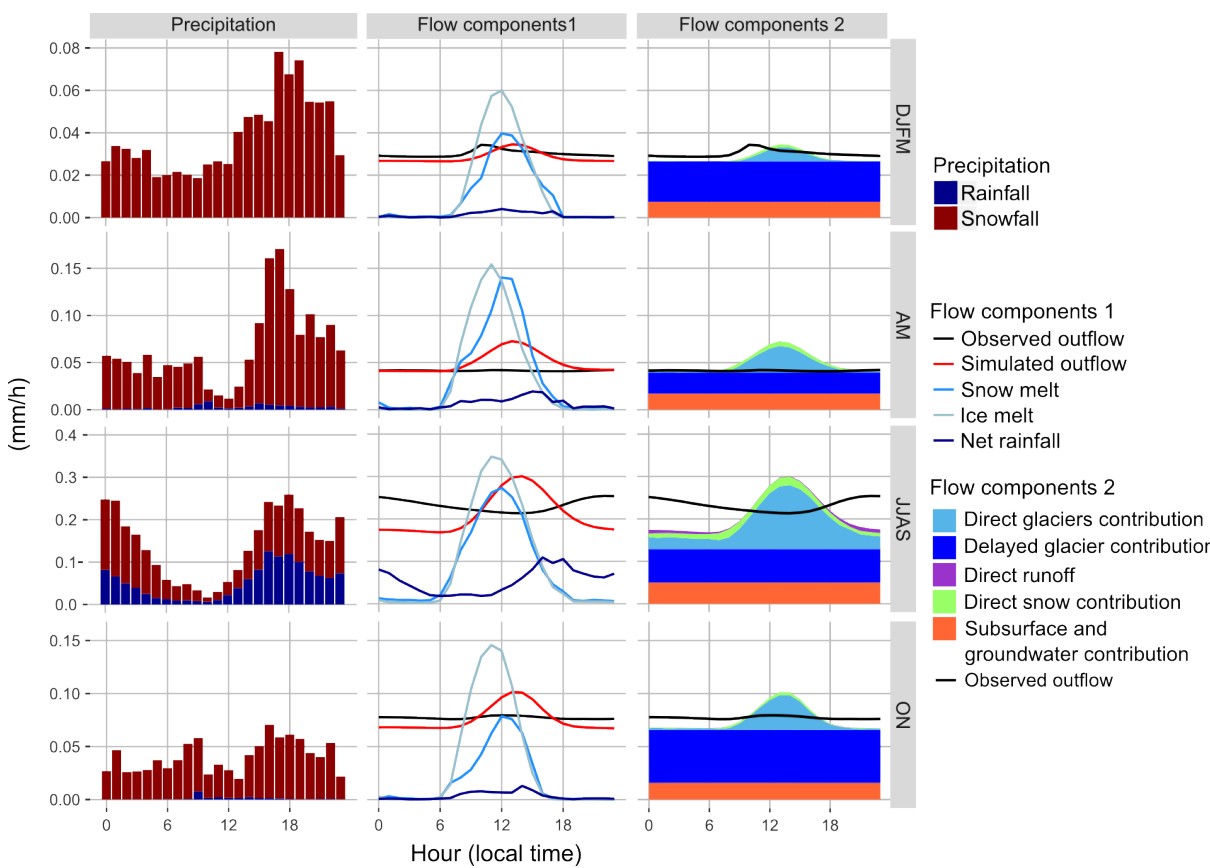

**Figure 15.** Mean hourly precipitation, discharge and flow components simulated with configuration v3 and averaged for the winter (DJFM), pre-monsoon (AM), monsoon (JJAS), and post-monsoon (ON) seasons. Note different y-axis scales for each season

# 5 Discussion

## 5.1 Simulation of the discharges and flow components

Soncini et al. (2016) studied flow components in the Pheriche catchment for the period 2013-2014 and estimated an annual ice melt component of 55 % and a snow melt component of 20 % of the annual outflow. The ice melt components are thus quite similar in terms of relative contributions to outflow, which is not the case for the snow melt components. We think that the main reason of such a difference is that we use different precipitation input. Indeed, precipitation data are measured here by Geonor sensors, while in Soncini et al. (2016) precipitation data come from tipping buckets. At the Pyramid station, where both sensors are installed, the Geonor sensor measures 60 % more precipitation than the tipping bucket over the period 2013-2015 and the main differences are in terms of solid precipitation (Mimeau et al., 2019).

Concerning the seasonal contributions to the outflow, our results are consistent with the results from Soncini et al. (2016), who found a main contribution of snow melt during the pre-monsoon season, mixed contributions of rainfall, snow melt and ice melt during the monsoon season and mixed contributions of snow melt and ice melt during post-monsoon and winter season. The studies of Ragettli et al. (2015) and Racoviteanu et al. (2013) concerning the Upper Langtang and the Dudh Koshi basin respectively, showed that most of the winter outflow surges from soil, channel, surface, and englacial storage changes, which is consistent with our results. However, the estimated flow components presented in this study, particularly the soil and englacial contributions, strongly depend on the model set-up. Figure 13 shows that the main part of the soil infiltrated water resurges within a day, whereas liquid water can be stored for several months within the glaciers. This difference between the response of the soil storage and the englacial storage results from the soil and glacier parameterization (see sensitivity analysis in Sect. 5.3.2).

At hourly scale, the results show that the model cannot represent the diurnal cycle of the outflow correctly as the simulated hydrological response is anticipated. Irvine-Fynn et al. (2017) found that on the Khumbu glacier the presence of supraglacial ponds buffers the runoff by storing diurnally more than 20 % of the discharge. This could explain the longer transfer time observed on the measured outflows which are not represented by the model. This shows that the current representation of the glacier storage in DHSVM-GDM does not allow to reproduce accurately the diurnal variations of discharge and further studies are needed in order to improve the model.

Overall, the comparison between the two definitions of the hydrological contributions shows that contributions must be explicitly specified in order to allow inter-comparison between models, especially for catchments with a large glacierized area. Moreover, the use of two different definitions allows to get complementary information on the origin of the outflow (processes at the origin of the runoff, types of flow generation, contributive zones). A perspective to improve the quantification of the hydrological contributions to the outflow is to track the icemelt, snowmelt and rainfall component pathways in the model as suggested in Weiler et al. (2018). This would enable to quantify the fractions of the three components contributing to subsurface and groundwater flow, which is not possible with the current version of DHSVM-GDM.

## 5.2 Representation of the cryospheric processes in the model

One of the main difficulties in glacio-hydrological modelling is to correctly simulate both river discharges, snow cover dynamics, and glacier mass balances. The results showed that two different representations of the cryospheric processes in the model (v0 and v3) can lead to similar simulated annual outflows but different estimations of the ice melt and snow melt contributions. This is particularly true for the glaciological year 2014-2015, when the ice melt contribution decreases from 59 % with v0 to 41 % with v3 and the snow melt contribution increases from 29 % to 47 % (Fig. 7). This can be explained by the fact that 2014-2015 was a year with a high amount of snowfall (Fujita et al., 2017), therefore, the representation of snow processes in the model has a larger impact on the simulated runoff production than the two other years. This highlights the importance of a correct representation of snow processes in the model. This also shows the need to use as much validation data as possible to assess the coherence between the ice, snow and hydrological processes and reduce the uncertainty on the flow components estimation.

The results also showed that the modification of one specific hydrological process (here, the representation of the snow albedo evolution) can have a significant impact on the simulated hydrological response of the catchment and requires improving other processes (here, considering specific representation of avalanches and debris-covered glaciers).

Further modifications of the model could also lead to different model results and it is also not excluded that different model errors are compensating each other. For example, the results showed that the original model version leads to a correct simulation of the river discharges because the non-representation of the insulation effect for debris covered glaciers on the ice melt was compensated by the incorrect representation of the snow albedo decrease. Due to the complexity of the model and the represented processes, no guarantee can be given that similar compensating effects still occur in the model. In this study, the validation of the model output was extended beyond the annual river discharge to discharges at different time scales, the snow cover area, and glacier mass balances in order to validate the simulations of the snow cover, glacier melt and discharges separately. The results demonstrate that the new version of the model performs well for all three signals. Moreover, the new parameterizations of the snow albedo and ice melt under debris were based on observed data (MODIS and in situ albedo measurements, and coefficient for ice melt under debris from Vincent et al. (2016)) and do not result from a calibration in order to avoid compensation effects. Therefore, it is very likely that the new implementation improved the quality of the represented processes.

The results presented in this study also indicate potential future improvements to increase the reliability and to reduce the uncertainty of the simulations at short time steps. While at daily and longer time scales the different hydrological components seem to be well reproduced by the model, the analysis of the diurnal cycle (Fig. 15) shows that DHSVM-GDM responds too rapidly to the ice melt production with too high diurnal peak discharges. This is probably related to the use of constant parameters in the parameterization of the englacial porous layer for glacier storage. Taking into account the seasonal variation of the efficiency of the englacial drainage system appears necessary to simulate the diurnal flow cycle correctly (Hannah and Gurnell, 2001). Therefore, further improvements should be based on studies analyzing the mechanisms of glacier drainage systems in the Khumbu region and their influence on glacier outflows (e.g., Gulley et al., 2009; Benn et al., 2017). These studies show that englacial conduits and supraglacial channels, ponds and lakes play a key role in the response of glaciers: DHSVM-GDM could thus be upgraded by implementing a parameterization of such systems and delay the response of glacierized areas, as successfully proposed, for instance, in the model developed by Flowers and Clarke (2002). Other processes such as supraglacial ponds and ice cliffs melting, transport of snow by wind or variation of temperature in the ice pack are not considered in DHSVM-GDM and their impact on the hydrological modelling should also be studied.

The avalanche routine implemented in this study is simplified and only considers 4 directions for the snow redistribution. A perspective of this study is too improve the representation of the avalanches in DHSVM-GDM by considering eight directions for the snow redistribution and considering other parameters such as the age of the snow cover, the snow density and the type of land-cover as it was proposed in Frey and Holzmann (2015).

| Glaciers inventory | Basin glacier area | | Glacier MB | Qsim | | Flow components (mm) | | |
|---|---|---|---|---|---|---|---|---|
| | km$^2$ | % | m w.e.yr$^{-1}$ | mm | Bias | Net rainfall | Snow melt | Ice melt |
| Racoviteanu et al. (2013) | 60 | 43 | -0.84 | 887 | -7% | 125 (13%) | 383 (41%) | 430 (46%) |
| GAMDAM | 38 | 24 | -1.17 | 824 | -13% | 123 (14%) | 387 (44%) | 370 (42%) |
| ICIMOD | 44 | 30 | -0.89 | 811 | -15% | 121 (14%) | 397 (46%) | 345 (40%) |

**Table 5.** Mean annual glaciers mass balance (MB), outflow and flow components simulated with different glaciers inventories (configuration v3)

## 5.3 Model sensitivities and uncertainties in forcing data

### 5.3.1 Sensitivity to the glacier outline

The three inventories used in this study result in very different estimates of the glacierized area: between 43 % and 24 % of the Pheriche basin with the inventories proposed by Racoviteanu et al. (2013) and GAMDAM (Fig. 3). Table 5 presents the average annual glacier mass balances, outflows, and flow components for the configuration v3 using the three inventories. The GAMDAM inventory leads to a more negative glacier mass balance than the two others inventories with -1.17 m w.e.yr$^{-1}$ compared to -0.84 and -0.89 m w.e.yr$^{-1}$ for the Racoviteanu et al. (2013) and ICIMOD inventories. This is due to smaller glacier accumulation areas in the GAMDAM inventory. The amount of snowfalls collected over those areas is lower, leading to more negative mass balances: glaciers receive less snowfall for accumulation, which lowers the mass balance value. Concerning the simulated outflow and flow components, the GAMDAM and ICIMOD inventories lead to fewer ice melt than the Racoviteanu et al. (2013) inventory due to their smaller areas in ablation zones, which leads to a smaller simulated annual outflow. From these results we estimate an uncertainty of 20 % (430 mm with the Racoviteanu et al. (2013) inventory versus 345 mm with the ICIMOD inventory, cf. Table 5) on the simulated annual ice melt volume related to the glaciers outline. The glacier outline mainly affects the simulated outflow during the monsoon season, when the ice melt contribution to the outflow is more important and leads to an uncertainty of 8 % (154 mm with the Racoviteanu et al. (2013) inventory versus 141 mm with the ICIMOD inventory) on the monthly discharges during monsoon season. This result shows that the choice of the glacier inventory as an input data of the glacio-hydrological model contributes to the uncertainty on the simulation results. Here, the Racoviteanu et al. (2013) inventory gives the best results in terms of glacier mass balance and the smallest bias with respect to the annual outflow. As its area is significantly larger than the other inventories, it gives the largest amount of ice melt. This potentially compensates a lack of precipitation due to a poor knowledge of the precipitation distribution over the catchment, specifically in the areas above 5000 m a.s.l. which constitute more than three quarters of the total area and for which no observations exist. It is worth noting that the glacier mass balances obtained with the Racoviteanu et al. (2013) and ICIMOD inventories are very similar but the amounts of simulated ice melt are different, which shows that a consistent mass balance can lead to errors on the simulated glacier contributions and total outflow.

### 5.3.2 Sensitivity to model parameter values

DHSVM-GDM is a physically based model with a large number of parameters which are difficult to define, particularly in a study area with little information on catchment characteristics and limited validation data. In order to have an indication of how sensitive the results are on the chosen parameter values, 27 additional simulations were performed with the new version of DHSVM-GDM (configuration v3). The model sensitivity concerning the melt coefficient on debris covered glaciers, the soil depth, the parameters of the englacial porous layer, and the avalanche parameters (see Table B1) was tested.

Figure 16 shows the impact of the parameter values on the simulated daily discharge. The overall variability ranges from 0.1 $m^3$/s up to 7.4 $m^3$/s, and is larger during the monsoon season (3.8 $m^3$/s on average in June). Overall, the parameter values have a low impact on the simulated daily discharge but a higher impact on the estimated relative contributions to the outflow (see Table B2). NSE values remain close to the reference simulation (around 0.91), while snow and ice melt contributions (definition 1) vary from 34 to 44 % and from 42 to 54 %, respectively. Concerning contributing areas (definition 2), the parameter values mainly impact the distribution between direct and delayed runoff from glacierized areas.

A major limitation to the estimation of the contributions from the runoff generation (overland flow, subsurface and ground-water flow, and englacial flow) is the representation of the englacial flows in the model. The performed simulations show that the parameter values of the porous englacial layer have a strong impact on the results (Table B2 sim. 3 to 11). While it has no impact on the total annual contributions from glacierized and non-glacierized areas, it impacts the partitioning between direct and delayed glacier contributions ranging from 34 to 20 % (direct) and 35 to 47 % (delayed), with the tested range of parameter values. The overall performance of the simulation of the daily discharge is affected by the choice of the parameter values of the porous englacial layer (NSE values ranging between 0.84 and 0.9). Indeed, an increase of the glacier storage capacity leads to a rather important delay of the outflow (as there is more infiltration simulated during the pre-monsoon and monsoon seasons). It is worth noting that we kept constant parameters for the englacial porous layer in our simulations, which is clearly a simplification, at least considering seasonally evolving flow networks within the glacier. A perspective would be to consider seasonal parameters values in order to reduce the uncertainty regarding direct and delayed runoff generation, but data to validate englacial storage and glacier outflows are currently not available, making it very difficult to adapt the parameter values to the study area.

Another limitation of our model lies in the application of a uniform reduction factor for ice melt under debris covered glaciers. Table B2 shows the sensitivity of the model to the ice melt reduction factor on debris covered glaciers (sim.1 and 2). When the reduction factor varies between 0.3 and 0.5, the simulated annual outflow is modified by $\pm7$ % and the mean ice melt flow component ranges from 42 to 50 %. In order to have a more realistic representation of the debris, the reduction factor could be spatially distributed, at least following elevation or slope exposition, and eventually taken as time-variant. As an example, Ragettli et al. (2015) considered a distributed debris thickness in their glacio-hydrological model and obtained a mean reduction of ice melt under debris of 84 %.

We also tested the model sensitivity with respect to the avalanche parameters by changing the values of the thresholds for the slope (35, 40, and 45 °) and the snow height (15, 30 and 60 cm) for steep terrains and for the snow water equivalent difference

between two neighboring cells (25, 50, and 100 cm) in case of a flat terrain (Tables B2 and B1 sim. 12 to 27). The results confirmed that higher thresholds for triggering avalanches and their propagation lead to less accumulation on the downslope parts of the glaciers and, thus, to more overall ice melt. Nevertheless, the results also show that the simulated outflow is not very sensitive to the avalanche parameterization since we derived a variation of 3 % of the mean annual outflow. However, the modification of the avalanche parameters leads to an uncertainty of 12 % on the estimation of the ice melt and snow melt components: the simulated contribution of ice melt to the water input varied between 46 and 52 % and the contribution of snow melt varied from 36 to 41 %.

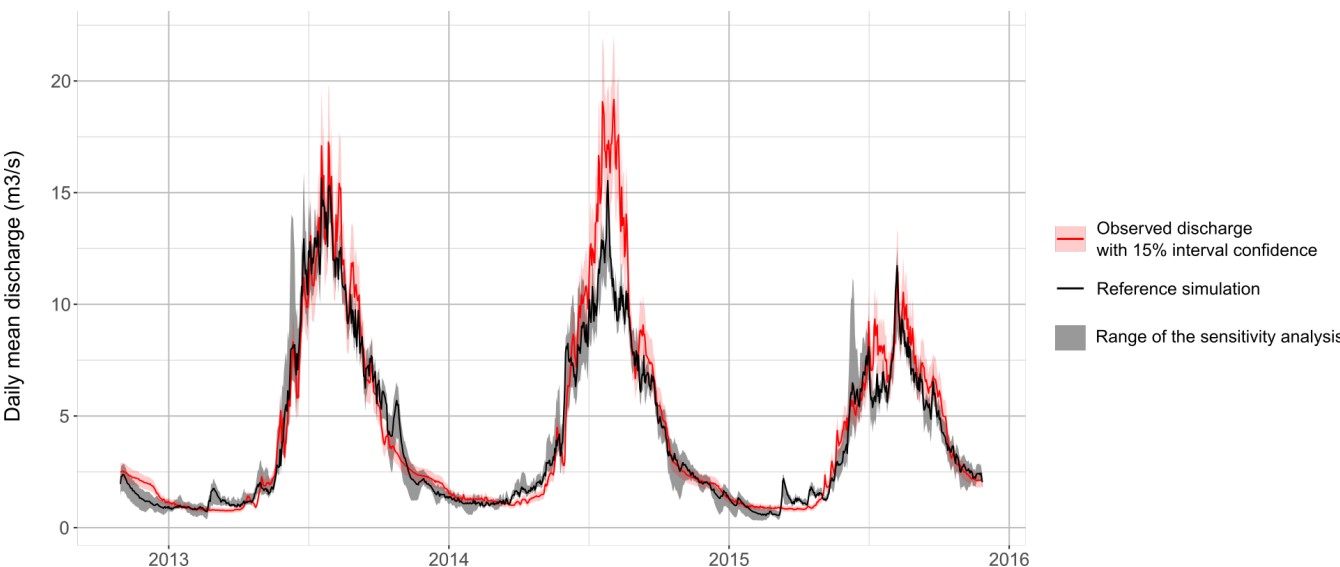

**Figure 16.** Sensitivity of the simulated daily discharge to different parameter values related to cryospheric processes. Black line represents the reference simulation (configuration v3), grey area shows the range (minimum and maximum values) of the 27 simulations with changing parameters (see Table B1), and observed discharges are represented in red.

### 5.3.3    Validation and forcing data uncertainties

A main issue is related to the availability of data for validating the glacio-hydrological modelling parameterizations and outputs. The lack of in-situ measurements at high elevations and the uncertainty related to the available data prevent from assessing the performance of the model in simulating the different cryospheric processes we consider. They only allow to evaluate integrated variables such as the annual glacier mass balance, the seasonal snow cover area dynamics and the outflow at the catchment outlet with significant uncertainties that impact the estimation of the different flow components. For instance, concerning the validation of the simulated glacier mass balances, only the order of magnitude of the simulated and geodetic mass balances used in this study can be compared because the considered areas are not the same in the different studies and the considered time periods differ as well. Indeed, four of the geodetic mass balances were derived for the Khumbu-Changri glacier, while mass balances from this study and from Brun et al. (2017) represent the mean mass balance for all glaciers located in the Pheriche

basin. Moreover, the mean annual glacier mass balance is estimated here for the three simulated years (2012-2015), whereas the geodetic mass balances correspond to longer (5 to 15 years) as well as earlier periods beginning between 1999 and 2002 (see Fig. 10) and do not take into account the inter-annual variability of the glacier mass balances. Nevertheless, on Figure 10, the variability of the simulated glacier mass balances is much larger than the variability of the geodetic mass balances, showing the significant impact of the snow and glacier parameterization on the simulation results. It is also worth noting that the snow cover distribution evaluation is particularly challenging on the Pheriche catchment, as clouds cover more than 50 % of the catchment during more than 150 days per year on average (and almost all the time during the monsoon season).

Finally, a major source of uncertainty lies in the lack of meteorological data at high elevation (because of the inaccessibility on the terrain) and in the measurement errors when observations are available, due to extreme meteorological conditions. In Mimeau et al. (2019), we tested the sensitivity of the model to different precipitation forcing data sets (in-situ, reanalysis, and satellite) and analyze the impact of different precipitation amounts and spatial distributions on the simulated discharges and flow components.

## 6   Conclusions

In this study we used a distributed physically-based glacio-hydrological model (DHSVM-GDM) to simulate the outflow of a small catchment in the Everest region and to estimate the different contributions to streamflows, which can be useful for water resources and water-related risks management. Different parameterizations of the snow and glacier processes in the model were compared in order to evaluate the uncertainty related to the representation of cryospheric processes on the simulated water balance. Simulated SCA were compared with MODIS images and calculated glacier mass balances with local in situ and geodetic measurements.

Results showed that the representation of the cryospheric processes in the model has a significant impact on the simulated outflow and flow components and that the improvement of one processes parameterization does not necessarily improve the overall simulation. Major contributions from glaciers and snow to the outflow were found, with 46 % of the annual water input produced by ice melt and 41 % by snow melt in the control simulation v3. Winter flows are mainly controlled by the release of englacial and soil water storage (up to 78 % in December in the control simulation v3), which corroborates other studies (Racoviteanu et al., 2013; Ragettli et al., 2015). These results are sensitive to both the chosen cryopheric parameterizations (v0 to v3 set-ups) and their relative parameter values (27 simulations added to the control v3), in a similar order of magnitude. For instance, we estimate an uncertainty related to the ice melt reduction factor by debris (ranging from 0.3 to 0.5) of $\pm 0.14$ m w.e.yr$^{-1}$ for the annual glacier mass balance and $\pm 7$ % for the annual outflow. Given the different tested parameter values, the uncertainty on the annual ice melt and the snow melt contributions ranges from -8 % to +17 % and from -17 % to +7 % with respect to the control contributions, respectively. When looking closely to the glacier contributions, these range from -23 to +38 % and from -20 to +14 % for the direct and delayed contributions, respectively. Finally,The glacier inventories used to outline the glacierized areas have also an important impact on the simulation results. The three inventories tested in

this study give estimations of the glacierized area ranging from 26 to 43 % of the basin area and a corresponding uncertainty of 20 % for the ice melt production.

This study also reminds that glacial and snow contributions to the streamflow must be clearly defined, as considering glacial contribution as the total outflow from the glacierized area or as outflow produced by ice melt can lead to very different estimations.

## Appendix A: Parameter values used in DHSVM-GDM

| Name | Unit | Value(s) | Reference |
|------|------|----------|-----------|
| Ground Roughness | m | 0.04 | Brutsaert (2005) |
| Reference Height | m | 2 | - |
| LAI Multiplier for rain interception | - | 0.0005 | Brutsaert (2005) |
| LAI Mulitplier for snow interception | - | 0.00005 | Andreadis et al. (2009) |
| Tree Height | m | 2 | - |
| Vegetation Density | - | 0.25 | - |
| Distance from bank to canopy | m | 2 | - |
| **Snow** | | | |
| Snow Roughness | m | 0.001 | Brock et al. (2006) |
| Rain Threshold | °C | 0 | L'hôte et al. (2005) |
| Snow Threshold | °C | 2 | L'hôte et al. (2005) |
| Snow Water Capacity | - | 0.05 | Singh (2001) |
| Minimum Intercepted snow | m | 0.005 | - |
| Maximum Snow Albedo | - | 0.85 | MODImLab |
| **Glaciers** | | | |
| Glacier Albedo | - | 0.3 | MODImLab |
| Melt coefficient for debris-covered glacier | - | 0.4 | Vincent et al. (2016) |

**Table A1.** Global parameter values used for the glacio-hydrological simulation with DHSVM-GDM

| Name | Unit | Value(s) | Reference |
|---|---|---|---|
| Soil Type | - | Soil (Regosol) | |
| Number of Soil Layers | - | 3 | - |
| Soil Lateral Conductivity | m/s | 0.0053 | Clapp and Hornberger (1978) |
| Soil Exponential Decrease | - | 2 | Niu et al. (2005) |
| Soil Depth Threshold | m | 10 | - |
| Soil Capillary Drive | - | 0.0756 | Morel-Seytoux and Nimmo (1999) |
| Soil Maximum Infiltration | m/s | 6.94E-06 | FAO |
| Soil Surface Albedo | - | 0.35 | ModimLab |
| Soil Porosity | - | 0.6 | calibrated |
| Soil Pore Size Distribution | - | 0.43 | Rawls et al. (1982) |
| Soil Bubbling Pressure | - | 0.302 | Rawls et al. (1982) |
| Soil Field Capacity | - | 0.31 | Meyer et al. (1997) |
| Soil Wilting Point | - | 0.23 | Meyer et al. (1997) |
| Soil Vertical Conductivity | m/s | 5.30E-05 | Clapp and Hornberger (1978) |
| Soil Thermal Conductivity layer 1 | W/m.°C | 7.114 | Burns (2012) |
| Soil Thermal Conductivity layer 2 et 3 | W/m.°C | 6.923 | Burns (2012) |
| Soil Thermal Capacity | J/m3.°C | 1.40E+06 | Burns (2012) |
| Number of Glacier Layers | - | 1 | - |
| Glacier Lateral Conductivity | m/s | 0.0003 | calibrated |
| Glacier Vertical Conductivity | m/s | 0.0003 | calibrated |
| Glacier Porosity | - | 0.8 | calibrated |

**Table A2.** Soil and glacier parameter values used for the glacio-hydrological simulation with DHSVM-GDM.

| Name | Unit | Value(s) | | | | Reference |
|---|---|---|---|---|---|---|
| Vegetation type | - | Shrubland | Grassland | Agriculture | Bare | |
| Overstory Present | - | FALSE | FALSE | FALSE | FALSE | - |
| Understory Present | - | TRUE | TRUE | TRUE | TRUE | - |
| Impervious Fraction | - | 0 | 0 | 0 | 0 | - |
| Height | m | 1 | 0.3 | 1 | 0.15 | - |
| Maximum Resistance | s/m | 600 | 600 | 600 | 600 | Wigmosta et al. (1994) |
| Minimum Resistance | s/m | 200 | 200 | 120 | 120 | Wigmosta et al. (1994) |
| Moisture Threshold | - | 0.6 | 0.6 | 0.33 | 0.8 | calibrated |
| Vapor Pressure Deficit | Pa | 2880 | 2880 | 4000 | 2000 | Wigmosta et al. (1994) |
| Rpc | - | 10 | 10 | 10 | 10 | Dickinson et al. (1991) |
| Number of Root Zones | - | 3 | 3 | 3 | 3 | - |
| Root Zone Depths 1 | m | 0.06 | 0.1 | 0.06 | 0.045 | - |
| Root Zone Depths 2 | m | 0.13 | 0.05 | 0.13 | 0.025 | - |
| Root Zone Depths 3 | m | 0.2 | 0.05 | 0.2 | 0.025 | - |
| Understory Root Fraction 1 | - | 0.4 | 0.4 | 0.4 | 0.4 | - |
| Understory Root Fraction 2 | - | 0.6 | 0.6 | 0.6 | 0.6 | - |
| Understory Root Fraction 3 | - | 0 | 0 | 0 | 0 | - |
| Understory Monthly LAI | - | 5.0 5.0 5.0<br>5.0 5.0 5.0<br>5.0 5.0 5.0<br>5.0 5.0 5.0 | 0.8 0.9 1.0<br>1.1 1.8 3.7<br>4.8 4.2 2.0<br>1.2 1.0 0.9 | 3.0 3.0 3.0<br>3.0 3.0 3.0<br>3.0 3.0 3.0<br>3.0 3.0 3.0 | 1.0 1.0 1.0<br>1.0 1.0 1.0<br>1.0 1.0 1.0<br>1.0 1.0 1.0 | Wigmosta et al. (1994) |
| Understory Albedo | - | 0.2 | 0.1 | 0.2 | 0.2 | Wigmosta et al. (1994) |

Table A3. Vegetation parameter values used for the glacio-hydrological simulation with DHSVM-GDM.

# Appendix B: Sensitivity analysis

| N° | Debris cov. gl. factor | Soil depth | Glacier porous layer | | | Hlim | Avalanches | | |
| | | | Depth | por | Ks | | slope | $\Delta H$ | Tr |
| | (-) | (m) | (m) | (-) | ($10^{-4}$ m/s) | (cm) | (°) | (cm) | (%) |
|---|---|---|---|---|---|---|---|---|---|
| Ref | 0.4 | 0.5 - 1.0 | 2 | 0.8 | 3 | 30 | 35 | 50 | 95 |
| 1 | **0.3** | 0.5 - 1.0 | 2 | 0.8 | 3 | 30 | 35 | 50 | 95 |
| 2 | **0.5** | 0.5 - 1.0 | 2 | 0.8 | 3 | 30 | 35 | 50 | 95 |
| 3 | 0.4 | 0.5 - 1.0 | **1** | 0.8 | 3 | 30 | 35 | 50 | 95 |
| 4 | 0.4 | 0.5 - 1.0 | **5** | 0.8 | 3 | 30 | 35 | 50 | 95 |
| 5 | 0.4 | 0.5 - 1.0 | 2 | **0.4** | 3 | 30 | 35 | 50 | 95 |
| 6 | 0.4 | 0.5 - 1.0 | 2 | **0.6** | 3 | 30 | 35 | 50 | 95 |
| 7 | 0.4 | 0.5 - 1.0 | 2 | **1** | 3 | 30 | 35 | 50 | 95 |
| 8 | 0.4 | 0.5 - 1.0 | 2 | 0.8 | **1** | 30 | 35 | 50 | 95 |
| 9 | 0.4 | 0.5 - 1.0 | 2 | 0.8 | **1.5** | 30 | 35 | 50 | 95 |
| 10 | 0.4 | 0.5 - 1.0 | 2 | 0.8 | **4.5** | 30 | 35 | 50 | 95 |
| 11 | 0.4 | 0.5 - 1.0 | 2 | 0.8 | **10** | 30 | 35 | 50 | 95 |
| 12 | 0.4 | 0.5 - 1.0 | 2 | 0.8 | 3 | **15** | 35 | 50 | 95 |
| 13 | 0.4 | 0.5 - 1.0 | 2 | 0.8 | 3 | **15** | **40** | 50 | 95 |
| 14 | 0.4 | 0.5 - 1.0 | 2 | 0.8 | 3 | **15** | **45** | 50 | 95 |
| 15 | 0.4 | 0.5 - 1.0 | 2 | 0.8 | 3 | 30 | **40** | 50 | 95 |
| 16 | 0.4 | 0.5 - 1.0 | 2 | 0.8 | 3 | 30 | **45** | 50 | 95 |
| 17 | 0.4 | 0.5 - 1.0 | 2 | 0.8 | 3 | **60** | 35 | 50 | 95 |
| 18 | 0.4 | 0.5 - 1.0 | 2 | 0.8 | 3 | **60** | **40** | 50 | 95 |
| 19 | 0.4 | 0.5 - 1.0 | 2 | 0.8 | 3 | **60** | **45** | 50 | 95 |
| 20 | 0.4 | 0.5 - 1.0 | 2 | 0.8 | 3 | 30 | 35 | **25** | **50** |
| 21 | 0.4 | 0.5 - 1.0 | 2 | 0.8 | 3 | 30 | 35 | **25** | **75** |
| 22 | 0.4 | 0.5 - 1.0 | 2 | 0.8 | 3 | 30 | 35 | **25** | 95 |
| 23 | 0.4 | 0.5 - 1.0 | 2 | 0.8 | 3 | 30 | 35 | 50 | **50** |
| 24 | 0.4 | 0.5 - 1.0 | 2 | 0.8 | 3 | 30 | 35 | 50 | **75** |
| 25 | 0.4 | 0.5 - 1.0 | 2 | 0.8 | 3 | 30 | 35 | **100** | **50** |
| 26 | 0.4 | 0.5 - 1.0 | 2 | 0.8 | 3 | 30 | 35 | **100** | **75** |
| 27 | 0.4 | 0.5 - 1.0 | 2 | 0.8 | 3 | 30 | 35 | **100** | 95 |

**Table B1.** Parameter values used for the sensitivity analysis. First line corresponds to the reference simulation (configuration v3) and simulations 1 to 27 refer to the sensitivity analysis simulations (parameters different from the reference simulation are highlighted).

| N° | NSE | Contributions Definition 1 (%) | | | Contributions Definition 2 (%) | | | | |
|----|-----|------------|-----------|----------|------|---------------|------|----------------|-----------------|
| | | **Net Rainfall** | **Snow Melt** | **Ice Melt** | Soil | Direct Runoff | Snow | Direct glacier | Delayed glacier |
| Ref | 0.91 | 13 | 41 | 46 | 24 | 2 | 6 | 26 | 43 |
| 1 | 0.88 | 14 | 44 | 42 | 26 | 2 | 6 | 24 | 42 |
| 2 | 0.92 | 12 | 39 | 49 | 23 | 2 | 5 | 28 | 43 |
| 3 | 0.88 | 13 | 41 | 46 | 24 | 2 | 6 | 34 | 34 |
| 4 | 0.88 | 13 | 41 | 46 | 24 | 2 | 6 | 20 | 49 |
| 5 | 0.84 | 13 | 40 | 47 | 24 | 2 | 5 | 28 | 42 |
| 6 | 0.89 | 13 | 40 | 47 | 24 | 2 | 5 | 27 | 43 |
| 7 | 0.9 | 13 | 40 | 47 | 24 | 2 | 5 | 26 | 44 |
| 8 | 0.89 | 13 | 40 | 47 | 24 | 2 | 5 | 36 | 34 |
| 9 | 0.9 | 13 | 40 | 47 | 24 | 2 | 5 | 32 | 38 |
| 10 | 0.9 | 13 | 40 | 47 | 24 | 2 | 5 | 24 | 46 |
| 11 | 0.89 | 13 | 40 | 47 | 24 | 2 | 5 | 21 | 49 |
| 12 | 0.91 | 13 | 40 | 47 | 24 | 2 | 5 | 26 | 44 |
| 13 | 0.91 | 13 | 40 | 47 | 24 | 2 | 5 | 26 | 43 |
| 14 | 0.9 | 13 | 39 | 48 | 24 | 2 | 5 | 26 | 43 |
| 15 | 0.9 | 13 | 40 | 47 | 24 | 2 | 5 | 26 | 43 |
| 16 | 0.9 | 13 | 39 | 48 | 24 | 2 | 5 | 27 | 43 |
| 17 | 0.9 | 13 | 40 | 47 | 24 | 2 | 5 | 26 | 43 |
| 18 | 0.9 | 13 | 39 | 48 | 24 | 2 | 5 | 26 | 43 |
| 19 | 0.9 | 13 | 38 | 48 | 24 | 2 | 5 | 27 | 43 |
| 20 | 0.9 | 13 | 37 | 50 | 24 | 2 | 5 | 26 | 43 |
| 21 | 0.9 | 13 | 39 | 48 | 24 | 2 | 5 | 26 | 43 |
| 22 | 0.91 | 13 | 41 | 46 | 24 | 2 | 6 | 26 | 43 |
| 23 | 0.9 | 13 | 35 | 52 | 24 | 1 | 5 | 27 | 44 |
| 24 | 0.9 | 13 | 36 | 51 | 24 | 1 | 5 | 26 | 44 |
| 25 | 0.9 | 13 | 34 | 54 | 23 | 1 | 5 | 27 | 44 |
| 26 | 0.9 | 13 | 35 | 53 | 23 | 1 | 5 | 27 | 44 |
| 27 | 0.9 | 13 | 36 | 51 | 23 | 2 | 5 | 27 | 44 |

**Table B2.** Nash coefficient (NSE) of simulated daily discharge and relative contributions to annual outflow (definitions 1 and 2) for the 27 sensitivity analysis simulations.

*Competing interests.* The authors declare that they have no conflict of interest.

*Acknowledgements.* This work has been supported by the French National Research Agency (ANR) through the ANR-13-SENV-0005-04/05-PRESHINE project and by a grant from Labex OSUG@2020 (investissement d'avenir. ANR 10 LABX56). It has benefited of data from the French Service d'Observation GLACIOCLIM (https://glacioclim.osug.fr/) and from Ev-K2-CNR (www.evk2cnr.org). The DHSVM-GDM model was made available by Chris Frans (Department of Civil and Environmental Engineering. University of Washington). The works of Yves Lejeune (CNRM-GAME) for correcting precipitation data in the GLACIOCLIM database and Ahmed Alatrash for processing MODIS data have been particularly helpful. Authors had fruitful discussions with Thomas Condom (IGE), Pierre Chevalier, Francois Delclaux, Luc Neppel and Judith Eeckman (HSM). Authors also acknowledge B. Schaefli, M. Weiler and a third anonymous reviewer for their comments and suggestions that significantly improved this paper content.

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
