# Peer review of "Quantification of different flow components in a high-altitude glacierized catchment (Dudh Koshi, Himalaya): some cryospheric-related issues"

_Hydrology and Earth System Sciences, 2018_

## Referee Comment (RC1) · Anonymous Referee #1 · 21 Feb 2018

This manuscript describes the effect of different model variants on model performance and simulated flow components. They perform various model tests using the physically-based DHSVM model. Especially the evaluation of flow components is a novel aspect. However, I am afraid that I have a list of rather major concerns as described in detail below. Major revisions, including new computations, are needed to bring this manuscript to its full potential.

Flow component definition

Defining and simulating flow components is not trivial and it is interesting that the authors here test different definitions. However, this discussion would be even more valuable if the authors could relate their definitions to those suggested by Weiler et al. (2018) (this reference is included, but not put in relation to the definitions used here).

Depending on the definition of flow components it can be required to track the different types of water through the model. For instance, glacier melt which is added to the groundwater (or 'soil', see below) might there mix with water coming from rain or snow melt. I am not fully sure, whether and, if yes, how this is done in DHSVM. Please clarify!

Model evaluation

I am missing a comparison of daily observed and simulated flows using a measure like the NSE. Why are these results not shown?

Given the uncertainties in observed snow cover and mass balances: are the differences in model performances really significant? Overall, I am missing a quantification of uncertainties (please see Pappenberger and Beven (2006)

Model parameterization

I am not convinced about the choice of the parameter values. Can one really use standard values for the parameters for this application in a high-mountain environment? The authors also state: "As a results, soil depth outside glacierized areas ranges between 0.5 and 1 m. Under the glaciers, the soil depth was set to 2 m." Why should we expect deeper soils under a glacier than elsewhere?!? This seems more like a trick to ensure a delayed response rather than a physically-based representation. As there is no description of groundwater, I assume that groundwater is not represented explicitly. The unrealistic soil depths are probably needed to compensate for the missing groundwater.

For the more sensitive parameters: how would changes in reasonable ranges affect results?

Please also address the devil's advocate question: the tested model variants might

just compensate differently for other model errors and if the rest of the model would be 'more correct' results would be different.

Avalanche routine

I have several concerns with this routine. First of all, while I agree with the authors on the need to consider snow redistribution, the way it is described it largely ignores other ways of snow redistribution than avalanches (see Freudiger et al, 2017, for a recent review on snow redistribution in hydrological models). Secondly, the routine and its parameterization seem rather ad hoc and not fully motivated. Can the parameters be motivated? What is the effect of varying them? Furthermore, as far as I understand the text, only cardinal directions are considered. Isn't that an unrealistic assumption? Also, from my understanding, I would assume that the avalanche routine would result in unrealistic line patterns of snow accumulation.

Finally, the avalanche routine is based on Wortmann et al. (2016). This, however, is a reference to a manuscript which had been in review (HESS-D) but then has never been published in HESS. I do not think we should refer to rejected manuscripts. This means a much better description (and motivation) of the routine is needed in this manuscript.

Structure and language

Please do not mix results and discussion. This makes reading the manuscript much more difficult. I strongly suggest separating these two sections.

There are a number of typos and places were grammar or words could be improved.

Freudiger, D., Kohn, I., Seibert, J., Stahl, K. and Weiler, M. (2017), Snow redistribution for the hydrological modeling of alpine catchments. WIREs Water, 4: n/a, e1232. doi:10.1002/wat2.1232

Pappenberger, F., and K. J. Beven (2006), Ignorance is bliss: Or seven reasons not to use uncertainty analysis, Water Resour. Res., 42, W05302, doi:10.1029/2005WR004820.

---

## Referee Comment (RC2) · B. Schaefli (Referee) · 7 Mar 2018

This manuscript proposes to quantify the origin of streamflow in a Himalayan basin, using a physically-based snow hydrological model. The underlying research question is interesting for the readership of HESS but I have the following major concern:

In this paper, two definitions of the origin of streamflow are used: A) annual contributions of snow fall, rainfall and ice melt to total runoff, and B) fractions of contributions coming from different areas. Both definitions can answer different questions and both are certainly useful. But my main question is: Is the water partioning and associated water flowpaths reliably enough represented in the used hydrological model to give reliable answers under definition A and B? What evidence do you have for such a reliable representation?

Based on the model description, I am not confident that this is the case. Delayed water release by glaciers is e.g. emulated with a deep soil under glaciers (as far as I understand), which does not necessarily give wrong results but the implications should be clearly discussed. Overall, the paper does not yet convincingly convey that the obtained results reliably represent the dominant hydrological processes. The paper validates snow and glacier mass balance simulations but no evidence is provided for a reliably parameterization of water partioning and release from the subsoil.

Detailed comments:

- Abstract: it is stated that "In general, it is shown that the choice of a given parametrization for the snow and glacier processes has a significant impact on the simulated water balance." Should there not be a more quantitative statement, including why the approach is nevertheless deemed useful to quantify the origin of water? I suggest mentioning all used validation data in the abstract (MODIS, mass balances)

- Introduction: it would be nice to better say why it is interesting to know the proportion of snow / ice melt and rainfall. One reason is that this can give insights into how much water is seasonally delayed and that this delay might change in the future. Another reason is that snow melt / ice melt might have a completely different hydrological pathway (in particular in terms of groundwater recharge) than rainfall. This might e.g. cause a shift in the overall water balance if the ratio snowfall to rainfall changes (Berghuijs Woods, Nature Climate Change, 2015). Another interesting question is how much water is currently available that has been accumulated long time ago in the glaciers.

- How does the model handle transpiration by vegetation? The loss via transpiration should be accounted for in the equations 4- 7 to quantify runoff production

- Results on winter flows controlled by release from the englacial water storage: what

are the similar results in the literature? What provides confidence that the model parameterization is reliable?

- Throughout the paper: what is net rainfall? There is no generally accepted definition.

- The cited observed geodetic mass balances have a very wide range of uncertainty and stem from different areas / different time periods. It is unclear why they are nevertheless useful for validating the modelling results. This should be justified. If the geodetic estimates are from a completely different period (period is not given), this might be questionable.

- Figure 10: I do not clearly see which model version is the best; in terms of RMSE, v3 might (slightly) outperform v0. What about the bias? Is it a good or a bad thing that v0 has less variability of the point mass balances than v3?

- Gauging curve uncertainty: what is the design of the gauge? Does the cross-section move? Is the uncertainty estimate not far too conservative? Please provide more details than "A 15

- General comment on conclusion: I strongly suggest to separate the discussion from the conclusion, it is very unusual to discuss results in the conclusion section

- Conclusion: can you really affirm that the model has an improved parameterization of the storage and transport of melt water within glaciers, or is the modified model just emulating it with the selected parameters? - Conclusion: instead of just stating that "The albedo parametrization (..) enabled to simulate the snow cover spatial distribution and the glacier mass balances more accurately", would be useful to refer to the validation data used

- Conclusion: "water is withdrawn every year from the catchment through ice melt"; strange formulation, difficult to understand; better something like "part of the streamflow leaving the catchment results from negative glacier mass balance changes".

- Conclusion: "Thus, if the precipitation regime (in terms of both intensity and phase)

[Figure]

does not change within the next decades, the access to water resources is likely to be reduced, especially during the fall and the winter seasons, as the glaciers outflow will decrease due to glaciers shrinkage, even without taking into account climate warming." This sentence should be deleted, it is pure guessing and perhaps wrong. Continued glacier retreat means continued negative mass balances, means water input in addition to annual precipitation. The moment of peak water remains to be determined.

- I am not a specialist in debris covered glaciers but I think that there should be some more literature review on how important a good representation of debris cover in glacio-hydrological models is, especially in the Himalaya

---

## Author Comment (AC1) · 19 May 2018

**Response to report # 1**

*This manuscript describes the effect of different model variants on model performance and simulated flow components. They perform various model tests using the physically-based DHSVM model. Especially the evaluation of flow components is a novel aspect.*
The authors of the manuscript thank the referee for his positive evaluation of the manuscript and for his comments, which helped improving the manuscript.

*However, I am afraid that I have a list of rather major concerns as described in detail below. Major revisions, including new computations, are needed to bring this manuscript to its full potential.*
Detailed responses to all components are given below (blue).

- *Flow component definition*

*Defining and simulating flow components is not trivial and it is interesting that the authors here test different definitions. However, this discussion would be even more valuable if the authors could relate their definitions to those suggested by Weiler et al.(2018) (this reference is included, but not put in relation to the definitions used here).*
The section 3.2.5 "Quantification of the flow components" has been modified in order to present the different types of contributions of the flow components as defined in Weiler et al. (2018) :

"Quantifying the relative contributions of ice melt, snow melt and rainfall in the river discharge at different time scales is a difficult task because hydrological models usually do not track the origin of water particles during transfer within the catchment (Weiler et al., 2018).
There are also different ways of defining the origins of the streamflow. Weiler et al. (2018) lists three types of contributions: 1) contributions from the source areas i.e. from each class of landcover , 2) contributions from the runoff generation (overland flow, subsurface flow, and groundwater flow), and 3) input contributions (icemelt, snowmelt, and rain).

In this study, two different definitions were used to estimate the hydrological contributions.
First, we estimate the input contributions to the total production of runoff (definition 1) according to the following equations […].

In order to evaluate the seasonal components of the outflow at the catchment's outlet, we also define the hydrological contributions as fractions of the outflow coming from the different contributive areas (definition 2) : […].

The definition 2 combines contributions from source areas (glacierized and non-glacierized areas) and contributions from runoff generation (direct runoff, englacial contribution, and soil contribution)."

*Depending on the definition of flow components it can be required to track the different types of water through the model. For instance, glacier melt which is added to the groundwater (or 'soil', see below) might there mix with water coming from rain or snow melt. I am not fully sure, whether and, if yes, how this is done in DHSVM. Please clarify!*
In DHSVM-GDM the amounts of icemelt, snowmelt and net rainfall are first estimated independently for each grid cell. Then, the three volumes are added up before estimating infiltration, runoff and losses by evapotranspiration. This means that the contributions corresponding to definition 2 are all a mix of icemelt, snowmelt and rainfall.
The following sentence was added to the section 3.2.5 "Quantification of the flow components" in order to clarify this point: "These contributions (definition2) are based on the simulated volume of water reaching the soil surface. This volume is a mix of icemelt, snowmelt and rainfall and can either infiltrate in the soil or produce runoff."

- *Model evaluation*

*I am missing a comparison of daily observed and simulated flows using a measure like the NSE. Why are these results not shown?*

NSE and KGE values were added to Figure 12. Furthermore, Table 3 with all calculated NSE and KGE values for the daily discharges for each model configuration was added to section 4.1.3 "Simulated Outflow and flow components".

|     | V0   | V1   | V2   | V3   |
| --- | ---- | ---- | ---- | ---- |
| NSE | 0.87 | 0.53 | 0.74 | 0.91 |
| KGE | 0.83 | 0.50 | 0.65 | 0.88 |

Table 3: NSE and KGE values calculated on the daily discharges on the period 2012-2015 for each model configuration.

Table 3 shows that configuration v0 leads to good simulated daily discharges with NSE and KGE values equal to 0.87 and 0.83. Although the implementation of the new parameterization of the snow albedo in configuration v1 improves the simulation of the SCA (Figure 8), it does not improve the simulation of the discharges since the values for NSE and KGE are reduced to 0.53 and 0.5. The implementation of the avalanche module in configuration v2 contributes to an improvement of the simulation of the daily discharges because the NSE and KGE values increase to 0.74 and 0.65. However, an improvement of the simulation with respect to the NSE and KGE values for the daily discharges is only reached with configuration v3 after the implementation of the new snow albedo parameterization, the avalanche module, and the reduction coefficient for the melt of debris covered glaciers leading to values of 0.91 and 0.88.

*Given the uncertainties in observed snow cover and mass balances: are the differences in model performances really significant? Overall, I am missing a quantification of uncertainties (please see Pappenberger and Beven (2006)*
In this study we do not have conducted a complete analysis of uncertainties of our simulations such that presented in Pappenberger and Beven (2006). The quantification of uncertainties is an analysis that would have required substantial work. Using a distributed model in such a task does demand very significant computational resources. Furthermore the existing DHSVM-GDM framework was not designed to support general uncertainty analysis.
We analysed the dependency in models factors (point 9.5) by using a multi-signal (snow, glacier and discharge) and a multiple evaluation criteria approach (NSE and KGE) to evaluate the quality of our results.
Our work could be considered as a contribution to the point 9.2 (Taking account of uncertainty in model choice), by giving a better documentation on the model development as proposed by Pappenberger and Beven (2006).
This study also analyses a large variety of sources of uncertainty by comparing results obtained with (i) different representations of the processes in the model (configurations v0, v1, v2, and v3), (ii) different parametrizations (melt coefficient for debris covered glaciers ranging from 0.3 to 0.5, soil depth under glaciers ranging between 1 and 5 m), (iii) different descriptions of the glacierized areas (Racoviteanu et al, 2013, GAMDAM and ICIMOD glacier inventories).
A second paper on the uncertainties associated to the precipitation forcing data (point 9.3) will be submitted soon (Mimeau et al., 2018). Overall, a large number of simulations were realized to analyze the uncertainties and the synthesis these simulations (presented in a thesis manuscript which will soon be published as well) shows that the main uncertainty is related to the precipitation forcing data.

- *Model parameterization*

*I am not convinced about the choice of the parameter values. Can one really use standard values for the parameters for this application in a high-mountain environment?*
The model DHSVM-GDM was developed to be used in high mountain environments and a lot of published results showed than these parameters were well adapted (Naz et al., 2014 ; Frans et al., 2015).
There is very little information about soil and vegetation properties in high-mountain environments. Since the vegetation in the study area is very limited and the soils are rather thin due to the steep topography, we selected standard values for the simulations. Moreover, data to validate subsurface and groundwater flows do not exist making it very difficult to adapt the parameter values to the study area. Obviously, the model parameters could have been calibrated based on the discharge measurements, but this method may have only compensated errors in other processes that are not well represented in the model (see below).

*The authors also state: "As a results, soil depth outside glacierized areas ranges between 0.5 and 1 m. Under the glaciers, the soil depth was set to 2 m." Why should we expect deeper soils under a glacier than elsewhere?!? This seems more like a trick to ensure a delayed response rather than a physically-based representation. As there is no description of groundwater, I assume that groundwater*

*is not represented explicitly. The unrealistic soil depths are probably needed to compensate for the missing groundwater.*

We agree with the referee that the 2 m soil depth under glaciers does not represent the real soil depth under glaciers. The selection of this value allowed a delay in the response for glacierized areas because in the glacier module icemelt is immediately transferred to the soil under the glaciers neglecting retention and storage of liquid water in the firn and ice..

The sentence in section 3.2.4 "Glaciers parametrization" referring to the soil depth under glaciers is modified to:

"In the original DHSVM-GDM version, glacier melt is instantaneously transferred to the soil, which is parameterized as bedrock under glaciers (Naz et al., 2014). This significantly underestimates the transfer time through glaciers. In this study we modified the soil parameterization for glacierized areas by increasing the soil depth to 2 m. This modification of the soil depth under glaciers compensates the absence of the representation of the retention and storage of liquid water in the firn and ice of the glaciers."

*For the more sensitive parameters: how would changes in reasonable ranges affect results?*

The soil parameterization does not impact the annual outflow and the annual contributions from glacierized and non-glacierized areas. However, it may impact the simulated seasonal discharges and the partitioning between direct and delayed contributions (soil water and englacial water contributions). We tested the sensitivity to the soil depth under glaciers (1, 2, and 5 m) with the configuration v3 (see supplementary information below). The increase of the soil depth leads to a slight delay of the outflow with very little impact on the NSE and KGE values. The soil depth under glaciers mainly impacts the estimation of the delayed contributions from glacierized area which ranges from 35 to 47 % of the total annual outflow when the soil depth varies between 1 and 5 m. A new section was added to the manuscript to present the sensitivity of the hydrological modelling to the soil depth under glaciers.

*Please also address the devil's advocate question: the tested model variants might just compensate differently for other model errors and if the rest of the model would be 'more correct' results would be different.*

Indeed, further modifications of the model will lead to different model results. It is also not excluded that different model errors are compensating each other. For example, our study demonstrated that the original model version leads to a correct simulation of the river discharges because the non-representation of the insulation effect for debris covered glaciers on the ice melt was compensated by the incorrect representation of the snow albedo decrease. Due to the complexity of the model and the resented processes, no guarantee can be given that similar compensating effects still occur in the model. Therefore, we extended the validation of the model output beyond the annual river discharge to discharges at different time scales, the snow cover area, and glacier mass balances in order to validate the simulations of the snow cover, glacier melt and discharges separately. The results demonstrate that the new version of the model performs well for all three signals. Moreover, the new parametrizations of the snow albedo and ice melt under debris were based on observed data (MODIS and in situ albedo measurements, and coefficient for ice melt under debris from Vincent et al. 2016) and do not result from a calibration in order to avoid compensation effects. Therefore, we are confident that the new implementation improved the quality of the represented processes and go well beyond a simple compensation of further modeling errors.

- *Avalanche routine*

*I have several concerns with this routine. First of all, while I agree with the authors on the need to consider snow redistribution, the way it is described it largely ignores other ways of snow redistribution than avalanches (see Freudiger et al, 2017, for a recent review on snow redistribution in hydrological models).*

*Secondly, the routine and its parameterization seem rather ad hoc and not fully motivated. Can the parameters be motivated? What is the effect of varying them?*

*Furthermore, as far as I understand the text, only cardinal directions are considered. Isn't that an unrealistic assumption?*

*Also, from my understanding, I would assume that the avalanche routine would result in unrealistic line patterns of snow accumulation.*

The glaciers located in the Pheriche catchment are known for being partially fed by avalanches, especially the West Changri Nup glacier whose mass balances are used in this study for validation (Sherpa et al, 2017). Ragettli et al, 2015 also showed the significant impact of avalanches on the river discharges in the Langtang region. In order to consider these processes in the model we implemented

a simplified avalanche module in the model and tested the sensitivity of the routine for the simulated water balance (snow cover, glacier mass balance and river discharge). The implementation of the avalanche routine led to an improvement on the simulated snow cover area and glacier mass balances compared to the versions with no representation of the snow redistribution and showed the importance of considering snow redistribution in glacio-hydrological models. Apparently, further potential improvements are possible: 1) considering 8 directions for the snow redistribution would indeed be more realistic and will necessitate the implementation of a more complex routine (in this study the avalanche routine is based on the overland flow routing algorithm of DHSVM-GDM which only considers 4 directions), 2) considering the redistribution of snow by wind. However, such modifications are beyond the scope of this study since they require more detailed data on the snow distribution based on in-situ observations for the study area. A sentence was added in the limitation section in order to present these potential model developments:
"The avalanche routine implemented in this study is simplified and only considers 4 directions for the snow redistribution. A perspective of this study is to improve the representation of the avalanches in DHSVM-GDM by considering 8 directions for the snow redistribution and considering other parameters such as the age of the snow cover, the snow density and the type of land-cover as it was proposed in Frey and Holzmann (2015)."

*Finally, the avalanche routine is based on Wortmann et al. (2016). This, however, is a reference to a manuscript which had been in review (HESS-D) but then has never been published in HESS. I do not think we should refer to rejected manuscripts. This means a much better description (and motivation) of the routine is needed in this manuscript.*
The avalanche module presented in Wortmann et al. (2016) (based on slope and snow height thresholds) seems realistic and the referees comments do not address this module, which is why we decided to cite this study even if it has not been published.

- *Structure and language*

*Please do not mix results and discussion. This makes reading the manuscript much more difficult. I strongly suggest separating these two sections.*
*There are a number of typos and places were grammar or words could be improved*
The structure of the manuscript was modified to separate the results and discussion sections.

**Quantification of different flow components in a high-altitude glacierized catchment (Dudh Koshi, Nepalese Himalaya). Supplementary information**

May 16, 2018

**1 Sensitivity of the soil depth under glaciers on the simulated discharges**

Three different values of soil depth under glaciers were tested in order to assess the sensitivity of the soil depth under glaciers on the simulated discharges. Figure 1 shows the simulated discharges and flow components simulated with configuration v3 for a soil depth under glaciers equals to 1 m, 2 m and 5 m.

Figure 1a shows that the soil depth under glaciers does not impact the simulated annual outflow and the total annual contributions from glacierized and non-glacierized areas. The soil depth under glaciers only impacts the partitioning between direct and delayed contributions (soil water and englacial water contributions): when the soil depth under glaciers ranges between 1 m and 5 m, the direct glaciers contribution ranges from 34 to 20 %, and the delayed glacier contribution ranges from 35 to 47 %.

An increasing of the soil depth under glaciers leads to a delay of the outflow as there is more infiltration simulated during the pre-monsoon and monsoon seasons, but this has a limited impact on the NSE and KGE values (see Figure 1b).

[Figure]

Figure 1: a) Annual and (b) monthly discharges and flow components (definition 2) simulated with configuration v3 with three different soil depths under glaciers (1 m, 2 m et 5 m). Observed monthly discharges are represented with an interval of uncertainty of 15 % (dashed back lines).

---

## Author Comment (AC2) · 19 May 2018

**Response to report # 2**

The authors of the manuscript thank the referee for his positive evaluation of the manuscript and for his comments, which helped improving the manuscript.

Detailed responses to all components are given below (blue).

*This manuscript proposes to quantify the origin of streamflow in a Himalayan basin, using a physically-based snow hydrological model.*

*The underlying research question is interesting for the readership of HESS but I have the following major concern:*

*In this paper, two definitions of the origin of streamflow are used: A) annual contributions of snow fall, rainfall and ice melt to total runoff, and B) fractions of contributions coming from different areas. Both definitions can answer different questions and both are certainly useful. But my main question is: Is the water partitioning and associated water flowpaths reliably enough represented in the used hydrological model to give reliable answers under definition A and B? What evidence do you have for such a reliable representation?*
*Based on the model description, I am not confident that this is the case.*

*Overall, the paper does not yet convincingly convey that the obtained results reliably represent the dominant hydrological processes. The paper validates snow and glacier mass balance simulations but no evidence is provided for a reliably parameterization of water partitioning and release from the subsoil.*
The validation of the model output was established on discharges at different time scales, the snow cover area, and glacier mass balances in order to validate the simulations of the snow cover, glacier melt and discharges separately. The results demonstrate that the new version of the model performs well for all three signals. Therefore, we are confident that the new implementation improved the quality of the represented processes and increases the reliability of the quantification of the ice melt, snow melt and rainfall contributions (definition 1).

There is no existing data to validate subsurface and groundwater flows making it difficult to assess the reliability of the contributions estimated with definition 2. Although these results are subject to a significant uncertainty, we think that addressing the question of the definition of the contributions and comparing the results obtained with two definitions is interesting for the glacio-hydrological community.

In order to present our approach more clearly, the "Results and discussion" section of the manuscript will be rearranged to:
- in a first part, present the contributions to the outflow simulated in the Pheriche catchment with an improved version of the glacio-hydrological model,
- in a second part, discuss all the limitations to the quantification of the contributions to the outflow (representation of the processes in the model, parametrization, initialization and water partitioning). In particular, a discussion concerning the uncertainty related to the partitioning between direct and delayed contributions will be added.

*Delayed water release by glaciers is e.g. emulated with a deep soil under glaciers (as far as I understand), which does not necessarily give wrong results but the implications should be clearly discussed.*
A test of sensitivity of the soil water depth under glaciers on the simulated hydrological response will be added in the manuscript. The results are presented in the supplementary information (below).

*Detailed comments:*

*- Abstract: it is stated that "In general, it is shown that the choice of a given parametrization for the snow and glacier processes has a significant impact on the simulated water balance." Should there not be a more quantitative statement, including why the approach is nevertheless deemed useful to quantify the origin of water?*

Replaced by: "The choice of a given parametrization for the snow and glacier processes has a significant impact on the simulated water balance: the different parametrizations tested in this study lead to an ice melt contribution to the outflow ranging from 45 to 70 %."

*I suggest mentioning all used validation data in the abstract (MODIS, mass balances)*
Sentence added in the abstract: "The validation of the snow, glacier and hydrological processes was established using three types of validation data (MODIS images, glacier mass balances and in situ discharge measurements)."

*- Introduction: it would be nice to better say why it is interesting to know the proportion of snow / ice melt and rainfall. One reason is that this can give insights into how much water is seasonally delayed and that this delay might change in the future. Another reason is that snow melt / ice melt might have a completely different hydrological pathway (in particular in terms of groundwater recharge) than rainfall. This might e.g. cause a shift in the overall water balance if the ratio snowfall to rainfall changes (Berghuij Woods, Nature Climate Change, 2015). Another interesting question is how much water is currently available that has been accumulated long time ago in the glaciers.*
The motivations for quantifying the contributions to the outflow should indeed be more explicit in the introduction. Thank you for these improvements which will be added in the introduction.

*- How does the model handle transpiration by vegetation? The loss via transpiration should be accounted for in the equations 4- 7 to quantify runoff production*
In DHSVM-GDM, the losses by evapotranspiration are withdrawn from the soil moisture. As there is no possibility of knowing the partitioning between ice melt, snow melt and rainfall in the soils simulated by the model, the contributions estimated with the definition 1 are calculated based on the volume of liquid water reaching the soil surface, i.e. before evapotranspiration.

Thank you for this remark, there is indeed an error in the equations 4 – 7 which was corrected as following:
```
Vrunoff = Vicemelt + Vsnowmelt + VrainNet
Vicemelt = VglAcc - Sice – dIwq/dt
Vsnowmelt = Psolid - Ssnow - VglAcc – dSwq/dt
VrainNet = Pliquid - Eint
Q = Vrunoff + ET
```

"Where VglAcc is the amount of snow that is transferred to the ice layer by compaction on glaciers (Naz et al., 2014), Sice and Ssnow are the amounts of sublimation from the ice and snow layers, dIwq/dt and dSwq/dt are the variations of the ice and snow storages, Psolid and Pliquid are the amounts of solid and liquid precipitations, and Eint is the amount of evaporation from intercepted water stored in the canopy.
It is worth noting that the sum of these contributions Vrunoff is not equal to the outflow at the catchment outlet Q as it represents all liquid water reaching the soil surface (before infiltration in the soils and glaciers and before evapotranspiration ET).
Moreover, at daily or monthly time scale, Vrunoff may not be equal to Q - ET as liquid water can be stored by or evacuated from the soil or glaciers."

*- Results on winter flows controlled by release from the englacial water storage: what are the similar results in the literature? What provides confidence that the model parameterization is reliable?*
Several studies have shown the role of the storage changes on the winter outflow in this region.
In the Langtang region, Ragettli et al. (2015) found that "storage changes (derived from changes in soil-, channel-, surface- and englacial reservoir volumes) are the most important contributors to runoff during winter".
Similar results were found by Racoviteanu et al. (2013): "Mixing models showed groundwater to be an important component of river flow within only tens of kilometers of the glacier outlets in the post-monsoon season".

With this model parametrization, it is difficult to separate groundwater and englacial flows, that's why the sentences p.21 l.10-13 will be replaced with:
"Figure 13b shows that groundwater and englacial water contributes to more than 50 % of the outflow during the monsoon season and can contribute up to 90 % during winter. This corresponds well to the studies of Ragettli et al. (2015) and Racoviteanu et al. (2013) concerning the Upper Langtang and the

Dudh Koshi basin respectively, who found that most of the winter outflow surges from soil and englacial storage changes."

As stated above, a discussion about the uncertainty on the estimation of the delayed contributions with DHSVM-GDM will be added in the revised manuscript.

*- Throughout the paper: what is net rainfall? There is no generally accepted definition.*
In this study, net rainfall is defined as the liquid precipitation minus the interception by the vegetation (equation 7).

*- The cited observed geodetic mass balances have a very wide range of uncertainty and stem from different areas / different time periods. It is unclear why they are nevertheless useful for validating the modelling results. This should be justified. If the geodetic estimates are from a completely different period (period is not given), this might be questionable.*
The in-situ punctual mass balances and the geodetic mass balances are used as complementary data for the glacier mass balances validation.
The geodetic mass balances are indeed estimated on different periods but these data give an order of magnitude of the glacier mass balances at the basin scale whereas the punctual mass balances only allow to validate the ice melt on the White Changri Nup and Pokalde glaciers, which are not representative of all glaciers in the basin.
Moreover, the White Changri Nup and Pokalde glaciers are two debris free glaciers and the punctual mass balances do not allow to validate the reduction factor on debris covered glaciers added in the model: on Figure 9, the comparison between the simulated glacier mass balances at the basin scale and the geodetic mass balances clearly shows that the configurations v1 and v2 lead to unrealistic glacier mass balances and that the debris layer on glaciers needs to be taken into account in order to simulate correct mass balances in the catchment.
For this type of highly glacierized catchments, using validation data from both local and global scales seems to be the best way to analyze the performance of the model for glacier melt simulation.

*- Figure 10: I do not clearly see which model version is the best; in terms of RMSE, v3 might (slightly) outperform v0. What about the bias? Is it a good or a bad thing that v0 has less variability of the point mass balances than v3?*
For the Pokalde glacier, the v0 and v3 RMSE values are similar but the configuration v3 leads to a bias ten times smaller than v0 (mean bias of 1 m with v0 and 0.1 m with v3).
V3 shows a larger variability but the point mass balances are spread around the diagonal axis. This is not the case with the configuration v0 which clearly always overestimates the mass balances.

*- Gauging curve uncertainty: what is the design of the gauge? Does the cross-section move? Is the uncertainty estimate not far too conservative? Please provide more details than "A 15*
The gauging station is located in gorges downstream from the village of Pheriche. The geometry of the measurement section was stationary. We did not observe any change of the channel cross section. The river stage was measured every 30 minutes with a pressure transducer (resolution -/+ 1 mm). 44 discharge measurements were performed using a tracer (fluorescein) dilution method (sudden injection). The tracer concentrations were measured each 5 seconds in the river downstream of the mixing zone using a fluorometer. A calibration was completed in the field for each gauging. This method is a powerful tool for measuring stream discharge, especially in steep, rough streams that cannot be gauged accurately using the velocity-area method (Hamilton and Moore, 2012). These measurements cover a range of water stages representing 95% of the range of variation observed during the study period.
The global uncertainty associated with a discharge time serie combines three main sources of errors: the uncertainties in the discharge measurement, in the measurement of stage, and in the plot of the stage-discharge relationship (Tomkins, 2012). In the case of natural channels it is difficult to predict this uncertainty precisely. In this study an estimation of its magnitude was proposed at 15%. This estimation combines the 3 sources of uncertainty: discharge measurements by dilution method (5 %); the uncertainties in stage measurement and time interpolation were considered as negligible and the uncertainty of the rating curve (10 %). These estimations were found in the literature (Di Baldassarre and Montanari, 2009; McMillan et al., 2012).

Di Baldassarre, G., Montanari, A., 2009. Uncertainty in river discharge observations: a quantitative analysis. Hydrol. Earth Syst. Sci., 13, 913–921. doi:10.5194/hess-13-913-2009

Hamilton, A.S., Moore, R.D., 2012. Quantifying Uncertainty in Streamflow Records. Canadian Water Resources Journal 37, 3–21. doi.org/10.4296/cwrj3701865

McMillan, H., Krueger, T., Freer, J., 2012. Benchmarking observational uncertainties for hydrology: rainfall, river discharge and water quality. Hydrological Processes 26, 4078–4111. doi.org/10.1002/hyp.9384

Tomkins, K.M., 2014. Uncertainty in streamflow rating curves: methods, controls and consequences. Hydrological Processes 28, 464–481. doi.org/10.1002/hyp.9567

*- General comment on conclusion: I strongly suggest to separate the discussion from the conclusion, it is very unusual to discuss results in the conclusion section*
The structure of the revised manuscript will be modified to separate the discussion and conclusions sections.

*- Conclusion: can you really affirm that the model has an improved parameterization of the storage and transport of melt water within glaciers, or is the modified model just emulating it with the selected parameters?*
There is indeed no representation of the storage and transport of melt water within glaciers in the modified version of the model as we used the soil parameterization to compensate this. We replace the following sentence:
"In this study, an improvement of the parameterization of cryospheric processes in DHSVM-GDM was proposed in order to better represent ice melt under debris covered glaciers, avalanches, the storage and transport of melt water within glaciers." (p.25 l.11-13)
with:
"In this study, an improvement of the parameterization of cryospheric processes in DHSVM-GDM was proposed in order to better represent ice melt under debris covered glaciers and avalanches"

*- Conclusion: instead of just stating that "The albedo parametrization (..) enabled to simulate the snow cover spatial distribution and the glacier mass balances more accurately", would be useful to refer to the validation data used*
Replaced with:
"Simulated SCA were compared with MODIS images and calculated glacier mass balances with local in situ measurements and geodetic mass balances. The parametrization proposed in this study and the implemented avalanche module enabled to simulate the snow cover spatial distribution and the glacier mass balances more accurately than in the original version of DHSVM-GDM by increasing the glacier accumulation and reducing ice melt." (p.25 l.14-16)

*- Conclusion: "water is withdrawn every year from the catchment through ice melt"; strange formulation, difficult to understand; better something like "part of the streamflow leaving the catchment results from negative glacier mass balance changes".*

*- Conclusion: "Thus, if the precipitation regime (in terms of both intensity and phase) does not change within the next decades, the access to water resources is likely to be reduced, especially during the fall and the winter seasons, as the glaciers outflow will decrease due to glaciers shrinkage, even without taking into account climate warming." This sentence should be deleted, it is pure guessing and perhaps wrong. Continued glacier retreat means continued negative mass balances, means water input in addition to annual precipitation. The moment of peak water remains to be determined.*
We agree to delete this sentence.

*- I am not a specialist in debris covered glaciers but I think that there should be some more literature review on how important a good representation of debris cover in glacio- hydrological models is, especially in the Himalaya*
Debris covered glaciers represent about 23 % of all glaciers in the Himalaya-Karakoram region (Scheler et al, 2011). The debris layers have been expanding during the last decades due to the glacier recession (Shukla et al, 2009, Bhambri et al, 2011, Benn et al, 2012) and are expected to keep expanding in the near future due to global warming (Rowan et al, 2015).

The debris thickness has a strong impact on the meltwater generation (Vincent et al, 2016 found a reduction factor of 0.4 between the ablation on debris covered areas and debris free areas on the

Changri Nup glacier) which means that a good representation of the debris covered glaciers in glacio-hydrological models is essential for estimating the right amount of meltwater generated in glacierized catchment in the Himalayas.

**Quantification of different flow components in a high-altitude glacierized catchment (Dudh Koshi, Nepalese Himalaya). Supplementary information**

May 16, 2018

**1 Sensitivity of the soil depth under glaciers on the simulated discharges**

Three different values of soil depth under glaciers were tested in order to assess the sensitivity of the soil depth under glaciers on the simulated discharges. Figure 1 shows the simulated discharges and flow components simulated with configuration v3 for a soil depth under glaciers equals to 1 m, 2 m and 5 m.

Figure 1a shows that the soil depth under glaciers does not impact the simulated annual outflow and the total annual contributions from glacierized and non-glacierized areas. The soil depth under glaciers only impacts the partitioning between direct and delayed contributions (soil water and englacial water contributions): when the soil depth under glaciers ranges between 1 m and 5 m, the direct glaciers contribution ranges from 34 to 20 %, and the delayed glacier contribution ranges from 35 to 47 %.

An increasing of the soil depth under glaciers leads to a delay of the outflow as there is more infiltration simulated during the pre-monsoon and monsoon seasons, but this has a limited impact on the NSE and KGE values (see Figure 1b).

[Figure]

Figure 1: a) Annual and (b) monthly discharges and flow components (definition 2) simulated with configuration v3 with three different soil depths under glaciers (1 m, 2 m et 5 m). Observed monthly discharges are represented with an interval of uncertainty of 15 % (dashed back lines).

---

## Author Response (AR1)

The modifications in the revised manuscript are given below (in blue) following the referees' comments (in black).
All line numbers in our replies refer to the revised version of the manuscript.

**Response to report # 1**

*This manuscript describes the effect of different model variants on model performance and simulated flow components. They perform various model tests using the physically-based DHSVM model. Especially the evaluation of flow components is a novel aspect.*
*However, I am afraid that I have a list of rather major concerns as described in detail below. Major revisions, including new computations, are needed to bring this manuscript to its full potential.*

- *Flow component definition*

*Defining and simulating flow components is not trivial and it is interesting that the authors here test different definitions. However, this discussion would be even more valuable if the authors could relate their definitions to those suggested by Weiler et al.(2018) (this reference is included, but not put in relation to the definitions used here).*
The section 3.2.5 "Quantification of the flow components" was modified in order to present the different types of contributions of the flow components as defined in Weiler et al. (2018):

p.10 l.12: "Quantifying the relative contributions of ice melt, snow melt, and rainfall in the river discharge at different time scales is a difficult task because hydrological models usually do not track the origin of water during transfer within the catchment (Weiler et al., 2018).
There are also different ways of defining the origins of the streamflow. Weiler et al. (2018) lists three types of contributions: 1) contributions from the source areas i.e. from each class of landcover, 2) contributions from the runoff generation (overland flow, subsurface flow, and groundwater flow), and 3) input contributions (ice melt, snowmelt, and rain).

p.10 l.17: In this study, two different definitions were used to estimate the hydrological contributions. First, we estimate the input contributions to the total production of runoff (definition 1) according to the following equations […].

p.11 l.1: In order to evaluate the seasonal components of the outflow at the catchment's outlet, we also define the hydrological contributions as fractions of the outflow coming from the different contributing areas (definition 2): […].

p.11 l.10: The definition 2 combines contributions from source areas (glacierized and non-glacierized areas) and contributions from runoff generation (direct runoff, englacial contribution, and soil contribution)."

*Depending on the definition of flow components it can be required to track the different types of water through the model. For instance, glacier melt which is added to the groundwater (or 'soil', see below) might there mix with water coming from rain or snow melt. I am not fully sure, whether and, if yes, how this is done in DHSVM. Please clarify!*
The following sentence was added to the section 3.2.5 "Quantification of the flow components" in order to clarify this point:
p.11 l.9: "These contributions are obtained from the amount of water reaching the soil surface simulated by DHSVM-GDM. On each grid cell, this volume is a mixture of ice melt, snowmelt and rainfall and can either infiltrate into the soil or produce runoff."

- *Model evaluation*

*I am missing a comparison of daily observed and simulated flows using a measure like the NSE. Why are these results not shown?*
p.16: Table 3 with all calculated NSE and KGE values for the daily discharges for each model configuration was added to section 4.1.3 "Simulated Outflow and flow components" (with comments p.15 l.21)

Furthermore, NSE and KGE values were added to Figure 13 (p.23).

*Given the uncertainties in observed snow cover and mass balances: are the differences in model performances really significant? Overall, I am missing a quantification of uncertainties (please see Pappenberger and Beven (2006)*

In this study we did not conduct a complete analysis of uncertainties of our simulations such as presented in Pappenberger and Beven (2006) since the existing DHSVM-GDM framework is not designed to support general uncertainty analysis. Instead, we analysed the dependency on model factors (point 9.5 according to Pappenberger and Beven, 2006) by using a multiple signal (snow, glacier and discharge) and criteria (NSE and KGE) approach to evaluate the quality of our results. Our work can be considered as a contribution to the point 9.2 (Taking account of uncertainty in model choice) as proposed by Pappenberger and Beven (2006) giving a better documentation of the model development.

This study also analyses a large variety of sources of uncertainty by comparing results obtained with (i) different representations of the processes in the model (configurations v0, v1, v2, and v3), (ii) different parametrizations (melt coefficient for debris-covered glaciers ranging from 0.3 to 0.5, soil depth under glaciers ranging between 1 and 5 m), and (iii) different descriptions of the glacierized areas (Racoviteanu et al, 2013, GAMDAM and ICIMOD glacier inventories).

Overall, a large number of simulations were realized to analyse the uncertainties. The synthesis of these simulations (presented in Mimeau, 2018) showed that the main uncertainty is related to the precipitation forcing data (point 9.3 according to Pappenberger and Beven, 2006). A manuscript on the uncertainties associated to the precipitation forcing data was recently submitted (Mimeau et al., 2018).

Mimeau, L., Quantification des contributions aux écoulements dans un bassin englacé par modélisation glacio-hydrologique. Application à un sous-bassin de la Dudh Koshi (Népal, Himalaya), Ph.D. thesis, University Grenoble Alpes, 2018

Mimeau , L., Esteves , M., Jacobi , H.W., Zin , I. [2018]. "Impact of precipitation uncertainty on the simulated hydrological response of a small glacierized Himalayan catchment". Submitted to Journal of Hydrometeorology.

- *Model parameterization*

*I am not convinced about the choice of the parameter values. Can one really use standard values for the parameters for this application in a high-mountain environment?*

*The authors also state: "As a results, soil depth outside glacierized areas ranges between 0.5 and 1 m. Under the glaciers, the soil depth was set to 2 m." Why should we expect deeper soils under a glacier than elsewhere?!? This seems more like a trick to ensure a delayed response rather than a physically-based representation. As there is no description of groundwater, I assume that groundwater is not represented explicitly. The unrealistic soil depths are probably needed to compensate for the missing groundwater.*

*For the more sensitive parameters: how would changes in reasonable ranges affect results?*

p.10 l.4: The sentence in section 3.2.4 "Glaciers parametrization" referring to the soil depth under glaciers was modified to:

"in the original DHSVM-GDM version, glacier melt is instantaneously transferred to the soil surface which is parameterized as bedrock under glaciers (Naz et al., 2014). This significantly underestimates the transfer time through glaciers. In this study we modified the soil parameterization in glacierized areas by increasing the soil depth to 2 m under glaciers. This modification of the soil depth under glaciers enables to compensate the absence of representation of the englacial liquid water storage in the model. This new parametrization also implies to change the values of three soil parameters under glaciers: the vertical and the lateral conductivities, as well as the porosity (Table A2). These were fixed by optimizing the recession shape of the hydrographs."

p.29: A sensitivity analysis of the parameters was added to the new section "5.3.2 Sensitivity to the soil and glacier parametrization"

*Please also address the devil's advocate question: the tested model variants might just compensate differently for other model errors and if the rest of the model would be 'more correct' results would be different.*

p.27 l6-16: A new paragraph was added discussing the possibility of the compensation of errors in the different model versions.

- *Avalanche routine*

*I have several concerns with this routine. First of all, while I agree with the authors on the need to consider snow redistribution, the way it is described it largely ignores other ways of snow redistribution than avalanches (see Freudiger et al, 2017, for a recent review on snow redistribution in hydrological models).*

*Secondly, the routine and its parameterization seem rather ad hoc and not fully motivated. Can the parameters be motivated? What is the effect of varying them?*

*Furthermore, as far as I understand the text, only cardinal directions are considered. Isn't that an unrealistic assumption?*

*Also, from my understanding, I would assume that the avalanche routine would result in unrealistic line patterns of snow accumulation.*

The glaciers located in the Pheriche catchment are known for being partially fed by avalanches, especially the West Changri Nup glacier, which are used in this study for the validation of mass balances (Sherpa et al, 2017). Ragettli et al, 2015 also showed the significant impact of avalanches on the river discharges in the Langtang region. In order to consider these processes in the model we implemented a simplified avalanche module in the model and tested the sensitivity of the routine for the simulated water balance (snow cover, glacier mass balance, and river discharge). The implementation of the avalanche routine led to an improvement of the simulated snow cover area and glacier mass balances compared to the versions without avalanches and confirmed the importance of considering snow redistribution in glacio-hydrological models. Apparently, further improvements are possible by: 1) considering eight directions for the snow redistribution would indeed be more realistic, but would need the implementation of a more complex routine (in this study the avalanche routine is based on the overland flow routing algorithm of DHSVM-GDM, which only considers 4 directions), 2) considering the redistribution of snow by wind. However, such modifications are beyond the scope of this study since they require more detailed data on the snow distribution based on in-situ observations for the study area. A sentence was added to section 5.2 'Representation of the cryospheric processes in the model' in order to present these potential model developments:

p.27 l.28: "The avalanche routine implemented in this study is simplified and only considers 4 directions for the snow redistribution. A perspective of this study is to improve the representation of the avalanches in DHSVM-GDM by considering eight directions for the snow redistribution and considering other parameters such as the age of the snow cover, the snow density and the type of land-cover as it was proposed in Frey and Holzmann (2015)."

*Finally, the avalanche routine is based on Wortmann et al. (2016). This, however, is a reference to a manuscript which had been in review (HESS-D) but then has never been published in HESS. I do not think we should refer to rejected manuscripts. This means a much better description (and motivation) of the routine is needed in this manuscript.*

The avalanche module presented in Wortmann et al. (2016) (based on slope and snow height thresholds) seems realistic and the referees' comments do not address this module, which is why we decided to cite this study although it has not been accepted for publication after the peer review, but it can be removed if needed.

- *Structure and language*

*Please do not mix results and discussion. This makes reading the manuscript much more difficult. I strongly suggest separating these two sections.*

*There are a number of typos and places were grammar or words could be improved*

The structure of the manuscript was modified to separate the results and discussion sections (see also our response to the first comment of referee # 2).

**Response to report # 2**

*This manuscript proposes to quantify the origin of streamflow in a Himalayan basin, using a physically-based snow hydrological model.*

*The underlying research question is interesting for the readership of HESS but I have the following major concern:*

*In this paper, two definitions of the origin of streamflow are used: A) annual contributions of snow fall, rainfall and ice melt to total runoff, and B) fractions of contributions coming from different areas. Both definitions can answer different questions and both are certainly useful. But my main question is: Is the*

*water partitioning and associated water flowpaths reliably enough represented in the used hydrological model to give reliable answers under definition A and B? What evidence do you have for such a reliable representation?*
*Based on the model description, I am not confident that this is the case.*

*Overall, the paper does not yet convincingly convey that the obtained results reliably represent the dominant hydrological processes. The paper validates snow and glacier mass balance simulations but no evidence is provided for a reliably parameterization of water partitioning and release from the subsoil.*

In order to present our approach more clearly, the "Results and discussion" section of the manuscript was rearranged in:
  - a first part, presenting the contributions to the outflow simulated in the Pheriche catchment with an improved version of the glacio-hydrological model (Section 4. Results p.15-25),
  - a second part, discussing all limitations for the quantification of the contributions to the outflow (representation of the processes in the model, parametrization, initialization and water partitioning). In particular, a discussion concerning the uncertainty related to the partitioning between direct and delayed contributions was added (Section 5. Discussion p.25-31).

*Delayed water release by glaciers is e.g. emulated with a deep soil under glaciers (as far as I understand), which does not necessarily give wrong results but the implications should be clearly discussed.*

A test of the sensitivity of the simulated hydrological response regarding the soil water depth under glaciers was added to section "5.3.2 Sensitivity to the soil and glaciers parametrization" p.29

*Detailed comments:*

*- Abstract: it is stated that "In general, it is shown that the choice of a given parametrization for the snow and glacier processes has a significant impact on the simulated water balance." Should there not be a more quantitative statement, including why the approach is nevertheless deemed useful to quantify the origin of water?*
*I suggest mentioning all used validation data in the abstract (MODIS, mass balances)*

The following information was added to the abstract:
p.1 l.15: "The choice of a given parametrization for the snow and glacier processes has a significant impact on the simulated water balance: the different parametrizations tested in this study lead to an ice melt contribution to the outflow ranging from 45 to 70 %."
p.1 l.11: "The validation of the snow, glacier and hydrological processes was established using three types of validation data (MODIS images, glacier mass balances and in situ discharge measurements)."

*- Introduction: it would be nice to better say why it is interesting to know the proportion of snow / ice melt and rainfall. One reason is that this can give insights into how much water is seasonally delayed and that this delay might change in the future. Another reason is that snow melt / ice melt might have a completely different hydrological pathway (in particular in terms of groundwater recharge) than rainfall. This might e.g. cause a shift in the overall water balance if the ratio snowfall to rainfall changes (Berghuij Woods, Nature Climate Change, 2015). Another interesting question is how much water is currently available that has been accumulated long time ago in the glaciers.*

The authors want to thank the referee for these suggestions, which were used to improve the introduction p.2 l.8-16.

*- How does the model handle transpiration by vegetation? The loss via transpiration should be accounted for in the equations 4- 7 to quantify runoff production*

In DHSVM-GDM, the losses by evapotranspiration are withdrawn from the overall soil moisture. Since there is no possibility of quantifying the partitioning between ice melt, snow melt, and rainfall in the soils simulated by the model, the contributions estimated with the definition 1 are calculated based on the volume of liquid water reaching the soil surface before evapotranspiration.

There is indeed an error in the equations 4 – 7, which was corrected as following:
Equation (8) was added p.10 to account for losses by evapotranspiration

p.10 l.28-31: "It is worth noting that the sum of these contributions Vrunoff is not equal to the outflow at the catchment outlet Q as it represents all liquid water reaching the soil surface (before infiltration in the soils and glaciers and before evapotranspiration ET).
Moreover, at daily or monthly time scale, Vrunoff may not be equal to Q - ET as liquid water can be stored by or evacuated from the soil or glaciers."

*- Results on winter flows controlled by release from the englacial water storage: what are the similar results in the literature? What provides confidence that the model parameterization is reliable?*
Since it is difficult to separate groundwater and englacial flows with the used model parametrization, the sentences p.23 l.2 were replaced by:
"Groundwater and englacial water represent a significant fraction of the monthly outflow as they contribute more than 50 % during the monsoon season and can contribute up to 90 % during winter (Figure 14b)."

A comparison of our results with other studies is included in the section "5.1 Simulation of the discharge and flow components":
p.26 l.12: "The studies of Ragettli et al. (2015) and Racoviteanu et al. (2013) concerning the Upper Langtang and the Dudh Koshi basin respectively, showed that most of the winter outflow surges from soil, channel, surface, and englacial storage changes, which is consistent with our results."

A discussion about the uncertainty on the estimation of the delayed contributions with DHSVM-GDM was added to the discussion (5.1 Simulation of the discharge and flow components and 5.3.2 Sensitivity to the soil and glaciers parametrization).

*- Throughout the paper: what is net rainfall? There is no generally accepted definition.*
In this study, net rainfall is defined as the liquid precipitation minus interception by the vegetation (equation 7 p.10).

*- The cited observed geodetic mass balances have a very wide range of uncertainty and stem from different areas / different time periods. It is unclear why they are nevertheless useful for validating the modelling results. This should be justified. If the geodetic estimates are from a completely different period (period is not given), this might be questionable.*
A new paragraph was added to section "5.3.3 Validation and forcing data uncertainties " (p.30 l.6 to p.31 l.4) to justify the use of the geodetic mass balances as well as the limits of such validating data.

*- Figure 10: I do not clearly see which model version is the best; in terms of RMSE, v3 might (slightly) outperform v0. What about the bias? Is it a good or a bad thing that v0 has less variability of the point mass balances than v3?*
The following sentence was added to the section "4.1.2 Snow cover dynamics":
p.19 l.10: "For the Pokalde glacier, the mass balances simulated with configuration v3 show a larger variability than the mass balances simulated with configuration v0, but the point mass balances are spread around the diagonal axis which leads to a bias ten times smaller (mean bias of 1 m with v0 and 0.1 m with v3)."

*- Gauging curve uncertainty: what is the design of the gauge? Does the cross-section move? Is the uncertainty estimate not far too conservative? Please provide more details than "A 15*
The gauging station is located in a gorge downstream from the village of Pheriche. The geometry of the measurement section was stationary. We did not observe any change of the channel cross section. The river stage was measured every 30 minutes with a pressure transducer (resolution ±1 mm). 44 discharge measurements were performed using a tracer (fluorescein) dilution method (sudden injection). The tracer concentrations were measured every five seconds in the river downstream of the mixing zone using a fluorometer. A calibration was completed in the field for each gauging. This method is a powerful tool for measuring stream discharge, especially in steep, rough streams that cannot be gauged accurately using the velocity-area method (Hamilton and Moore, 2012). These measurements cover a range of water stages representing 95% of the range of variation observed during the study period.
The global uncertainty associated with a discharge time series combines three main sources of errors: the uncertainties in the discharge measurement, in the measurement of the stage, and in the plot of the stage-discharge relationship (Tomkins, 2012). In the case of natural channels it is difficult to predict this uncertainty precisely. In this study an estimation of its magnitude was proposed at 15%.

This estimation combines the 3 sources of uncertainty (Di Baldassarre and Montanari, 2009; McMillan et al., 2012): discharge measurements by the dilution method (5 %); the uncertainties in stage measurement and time interpolation (negligible), and the uncertainty of the rating curve (10 %).
Di Baldassarre, G., Montanari, A., 2009. Uncertainty in river discharge observations: a quantitative analysis. Hydrol. Earth Syst. Sci., 13, 913–921. doi:10.5194/hess-13-913-2009

Hamilton, A.S., Moore, R.D., 2012. Quantifying Uncertainty in Streamflow Records. Canadian Water Resources Journal 37, 3–21. doi.org/10.4296/cwrj3701865

McMillan, H., Krueger, T., Freer, J., 2012. Benchmarking observational uncertainties for hydrology: rainfall, river discharge and water quality. Hydrological Processes 26, 4078–4111. doi.org/10.1002/hyp.9384

Tomkins, K.M., 2014. Uncertainty in streamflow rating curves: methods, controls and consequences. Hydrological Processes 28, 464–481. doi.org/10.1002/hyp.9567

*- General comment on conclusion: I strongly suggest to separate the discussion from the conclusion, it is very unusual to discuss results in the conclusion section*
The structure of the revised manuscript was modified to separate the discussion and conclusions sections.

*- Conclusion: can you really affirm that the model has an improved parameterization of the storage and transport of melt water within glaciers, or is the modified model just emulating it with the selected parameters?*
In this study, we indeed did not modify the parametrization of the storage and transport of melt water within glaciers. Therefore, we replaced the following sentence:
"In this study, an improvement of the parameterization of cryospheric processes in DHSVM-GDM was proposed in order to better represent ice melt under debris covered glaciers, avalanches, the storage and transport of melt water within glaciers."
with:
"Some improvements on the cryospheric processes parameterization in DHSVM-GDM were proposed in order to better represent the snow cover dynamics, the ice melt under debris-covered glaciers, and avalanches." (p.31 l.18)

*- Conclusion: instead of just stating that "The albedo parametrization (..) enabled to simulate the snow cover spatial distribution and the glacier mass balances more accurately", would be useful to refer to the validation data used*
The following sentence was added to the conclusion:
p.31 l.20: "Simulated SCA were compared with MODIS images and calculated glacier mass balances with local in situ and geodetic measurements".

*- Conclusion: "water is withdrawn every year from the catchment through ice melt"; strange formulation, difficult to understand; better something like "part of the streamflow leaving the catchment results from negative glacier mass balance changes".*
*- Conclusion: "Thus, if the precipitation regime (in terms of both intensity and phase) does not change within the next decades, the access to water resources is likely to be reduced, especially during the fall and the winter seasons, as the glaciers outflow will decrease due to glaciers shrinkage, even without taking into account climate warming." This sentence should be deleted, it is pure guessing and perhaps wrong. Continued glacier retreat means continued negative mass balances, means water input in addition to annual precipitation. The moment of peak water remains to be determined.*
These sentences were deleted.

*- I am not a specialist in debris covered glaciers but I think that there should be some more literature review on how important a good representation of debris cover in glacio- hydrological models is, especially in the Himalaya*
The following sentences were added to the introduction:

[revised manuscript text omitted]

---

## Author Response (AR2)

The authors addressed the reviewers' comments well overall. The paper could, however, still gain from using a more precise terminology (examples below).

The authors thank Bettina Schaeffli for the positive comments and advice, which helped to improve the manuscript. Detailed responses to all comments are given below (blue).

Detailed comments:

- "Definition 1 allows assessing the annual impact of glaciers and snow cover (…)". Correct would probably be " (..) impact of glacier melt and snow melt".

The sentence was corrected to:

p.11 l.11-12: "Definition 1 allows assessing the annual impact of glacier melt and snow melt on the water production".

- Section 4.1.3 does still not indicate for which period the geodetic mass balances have been estimated; it is only in the figure legend; I suggest adding a note in the text

The periods for the geodetic mass balances were added to section 4.1.3:

p.18 l.1-5: "Figure 10 compares the simulated mean annual glacier mass balances obtained with the different model configurations with mass balances determined with geodetic methods between 1999 and 2015 (Bolch et al., 2012; Gardelle et al., 2013; Nuimura et al., 2015; King et al., 2016; Brun et al., 2017). These geodetic mass balances range from -0.67 ± 0.45 m w.e.yr -1 (2000-2008) (Nuimura et al., 2015) to -0.32 ± 0.09 m w.e.yr-1 (2000-2015) (Brun et al., 2017)."

- Even if your definitions are correct, I would still be as precise as possible everywhere. Thus the sentence should read "of the outflow as coming from water infiltrated in glaciers and more than 20% from subsurface and groundwater flows (NEW:) generated outside the glacier covered area"

The sentence was modified as suggested (P.20 l.21)

- Figure 13: I would remove the discharge from the top figure since the figure mixes inputs (melt etc.) and outputs (streamflow) and it wrongly implies that you can label how much water comes from melt at a certain time step (which your approach cannot)

We modified Figure 13 and removed the discharge from the top panel.

- Terminology:

- "Total production of runoff" is misleading since this is not water that creates runoff but the water available in the system (the definition is correct but I suggest changing the terminology).

"On average, we find that the outflow is mainly produced by meltwater (…)"; this certainly holds but in fact, you assess the input from these sources, not the outflow. The same problem repeats in the conclusion: "major contributions from glaciers and snow (…) of the annual outflow produced by ice melt".

Figure 14: here it is extremely obvious that a renaming of "runoff production" is necessary (see later): someone who reads figure 14, necessarily thinks that you assessed how much runoff comes from melt and rain. Why not call it "Total water input"?

We changed the terminology and replaced "runoff production" by "water input" in the text, tables and figures.

For example, p.20 l.16-17: "On average, we find that the outflow is mainly produced by meltwater as 46 % of the annual water input is due to ice melt and 41 % to snow melt (definition 1)."

- V stands usually for Volume; it is however a flux. I suggest changing the symbol and give the units [L/T]

Equation 8 should read: dStorage/dt= V(t) – Q(t) – ET(t), only if dS/dt (in the catchment) is zero, we have Q(t) = V(t) – ET (t) (ET is subtracted not added). This avoids saying that dS/dt might not be zero. Writing all previous equations as storage change = input – output would probably avoid confusion

Thank you for pointing out this error in the equation. Section "3.2.5 Quantification of the flow components" was modified accordingly

- Italic is only used for variables, not for standard functions such as "max"; units are not in italic;

- Textual subscripts or superscripts should not be italic (e.g. xmax, Tmin where "max" and "min" stand for maximum and minimum, respectively).

The equations of the sections 3.2.2 and 3.2.5 were modified to take into account the two previous remarks.

Detailed responses to all comments are given below (blue).

While the manuscript certainly has been improved, I am a bit disappointed that the authors focused rather on the 'simple' revisions than the more substantial ones. As stated in my previous review, I think that model changes and new computations would be needed. To repeat, my concerns regarding the snow redistribution remain:

1) Only avalanches as redistribution

2) Only cardinal directions

We agree that there are limitations regarding the snow redistribution in our model and we addressed the concerns of the reviewer already during the discussion phase. In short, we concede that a more sophisticated module has the potential to better reflect real conditions for the redistribution of snow. However, processes driving this redistribution like blowing snow require detailed input data concerning the spatially distributed wind direction and speed (Freudiger et al, 2017). Wind fields for the entire catchment derived with the available data (wind speed was measured at two stations with substantial gaps in the data series) would be prone to large and potentially systematic errors. Therefore, we decided to implement at this stage only a simplified module for the snow redistribution, which is also coherent with the current overland flow routing algorithm of DHSVM-GDM considering four directions (Wigmosta et al., 1994). We are well aware of these limitations, which are, thus, discussed in detail in the section "5.2 Representation of the cryospheric processes in the model".

3) Description in a rejected manuscript

We removed the citation from the manuscript.

1) The same applies to the issue of literature values, ad hoc decisions and 'tricks', such as ... My concern regarding lit values has not been addressed

The referee asked for a sensitivity analysis for the most sensitive model parameters. While the objective of this study was certainly not to perform a full sensitivity analysis of all model parameters for a well-established model like DHSVM, we performed sensitivity analyses concerning the newly implemented cryospheric processes concerning for example the glacier outline, the melt coefficient for debris-covered glaciers, and the avalanche parameters. Furthermore, the model sensitivity to the soil parameterization was analyzed. Due to the sparse vegetation and the dominating cryospheric processes in the study catchment we believe that these are the most parameters for a sensitivity analysis for our model set-up. The results of the sensitivity analysis are presented in chapter 5.3.

2) 2 m soil under glaciers is just unrealistic and I find it unfortunate that in a manuscript which aims at more realistic simulations such tricks are used to compensate for missing processes in the model. Honestly, I am also not fully happy by the uncertainty analyses.

It is unclear to us to why the referee refers to "tricks" concerning our manuscript. Obviously, any model is an imperfect representation of the real world. Is using a certain parameterization or certain values for model parameters a "trick"? Unfortunately, not all information one would like to have for the selected catchment is available. Nevertheless, we believe that doing simulations for such a region

brings new results for the scientific community and advances our understanding. If observations were not available we performed simulations with parameter values according to the best of our knowledge. These choices are described in the manuscript. While the referee obviously disagrees with (some of) our choices, we do not understand what justifies to qualify these choices as "tricks". Concerning the soil depth we refer to the study of Pelletier et al. (Global 1-km Gridded Thickness of Soil, Regolith, and Sedimentary Deposit Layers. ORNL DAAC, Oak Ridge, Tennessee, USA. http://dx.doi.org/10.3334/ORNLDAAC/1304, 2016), which indicates for more than 50% of the Pheriche catchment area soil depths ranging between 1 and 2 m. While it remains unclear why the referee claims that a soil depth of 2 m, which is used for most model runs, is "unrealistic", we agree that the soil depth is highly uncertain especially for areas with glaciers. Therefore, we tested the model sensitivity with respect to the soil depth. The results are summarized in chapter 5.3.2.

As a minor comment, I also find the refs to a thesis in French and a submitted manuscript not convincing.

These references are not used in the manuscript and were only used in our response to the referee in order to illustrate other aspects of our work concerning the characterization of the uncertainties in our study. Although written partly in French, the thesis has been evaluated and reviewed by a Ph.D. jury and is published here: https://tel.archives-ouvertes.fr/tel-01863806.  The manuscript is currently under revision after submission to the Journal of Hydrometeorology in July 2018 (Mimeau et al., Impact of precipitation uncertainty on the simulated hydrological response of a small glacierized Himalayan catchment, JHM-D-18-0157).

Perhaps I am missing something, but I still do not fully understand how the flow components were tracked. Actually, I am now even more confused than before. Did the authors change things in the DHSVM code? If yes, what? If no, how can then the components be tracked with the standard DHSVM?

We only changed the code to have access to internally simulated variables, which are not routinely included in the output of the standard version of DHSVM. No changes were made to the calculation of these variables in the model.

Concerning definition 1, flow components were computed using the water balance equations 5 to 8described in section "3.2.5 Quantification of the flow components". All variables used for the computation are part of standard output or internal variables of DHSVM.

Concerning the contributions estimated with definition 2, the method used for the computation was added in a supplementary material.

[revised manuscript text omitted]

---

## Author Response (AR3)

Editor report:
Dear authors,

as you can see, the referee report still asks several valid questions. I took the time to have a more detailed look than normal to the revised paper and the your responses. I also believe, that there are several issues with the paper, that need to be fixed before the paper could be accepted for publication in HESS. I know, some of them are not easy, but being partly a field hydrologist myself, I cannot accept that we claim to develop better models but at the end, we just use some simple fixes like adding a soil storage underneath a glacier to represent the much more complex process of englacial liquid water storage, where conceptual models have already been developed. When looking in more detail into the sensitivity analysis in chapter 5.3, I was also wondering about the presented NSE and KGE values for changing the "soil depth" under the glacier (at least you should give this a different name!!) and the values in Table 3. Why are some of the values higher than the best model in Table 3. The sensitivity of changing this parameter is actually larger than the improvement of the model from v0 to v3. If this is the case, why improving the other parts if this process would be so much more relevant to improve streamflow prediction.
From the three points proposed by the referee, I would at least see that the following points should be addressed:
• Explicitly adding storage in the glacier (see above), which might reduce the need to add a thick soil under the glacier
• Performing a limited number of Monte Carlo runs with reasonable parameter value ranges - as you clearly demonstrate, there are many uncertainties and then you parameterise some parameters by "These were fixed by optimizing the recession shape of the hydrographs." This is not repeatable by other scientists.

I hope you see the need to reconsider some parts of your model and to change the analysis and provide a clear way to explain the model and its implications.

Referee report:
The authors seem a bit upset by the term 'trick'. However, I think we need to honest about our modeling approaches. As I understand the parameterization choice, the soil below the glacier is assumed to be 2m and, thus, thicker than the soils in the rest of the catchment. I am sorry, but this is not physically realistic but a 'trick' to compensate for other processes that are apparently missing in the model.
Re snow-redistribution: I fully agree with the authors, that there is almost no limit on how detailed one could go and this would be beyond this study. However, having said this, I still think that the approach being used here is overly simplistic. At least allowing for diagonal transport downslope would seem like an easy improvement. Btw, it is still not clear to me, how the snow is distributed among downslope cells in case of several downslope directions. To consider the uncertainties, especially also those arising from using literature values, which were partly derived from rather different geographical settings, I would still like to see some Monte Carlo / ensemble approach. Even a small number of model runs with different values for all parameters would at least give an indication of how sensitive the results are to the particular parametrization.
The simple solution would be to clearly discuss these limitations in the text (and in a more prominent way than the uncertainty analyses now, which comes a bit hidden in the end).

However, I would encourage the authors to consider:
• Adding diagonal snow redistribution
• Explicitly adding storage in the glacier, which might reduce the need to add a thick soil under the glacier
• Performing a limited number of Monte Carlo runs with reasonable parameter value ranges

Authors response:

Dear Editor,

Please find herewith a modified version of our paper "Quantification of different flow components in a high-altitude glacierized catchment (Dudh Koshi, Himalaya)". The authors thank the editor Markus Weiler and the anonymous referee for their advices on the glacier parameterization and sensitivity analysis, which undeniably helped to improve the manuscript.

The main modifications that were applied to the manuscript are the following:

The abstract, the discussion and conclusion of the manuscript were modified to stress on the analysis of the impact of the representation of cryospheric processes on the simulated hydrological response, rather than on the potential improvements of the latest version of the model. Indeed, the main objective of the manuscript is not to propose a new model, but to discuss on the difficulty to correctly represent cryospheric processes in glacio-hydrological models (especially in a scarce data region as the Himalayas) and to assess the uncertainty on the relative contributions of ice and snow melt to the streamflow using such models.
• Englacial storage parameterization
Instead of using an additional soil depth under glaciers as in the initial version of the paper, we now consider a conceptual englacial porous layer between the glacier and the bedrock for taking account of the melt storage and the consequent delay of the snow and ice melt.
This alternative conceptual parameterization is probably more adapted to a physically based model as DHSVM, than a classical conceptual reservoir. Of course, one has to fix the porous layer parameter values, i.e. the depth of the englacial layer, as well as the porosity and the hydraulic conductivityand on this specific point, we would like to clarify the approach that we followed. As we do not have flow or storage measurements within the glacier, but we do have observed hydrographs at the outlet, instead of just optimizing the global NSE or KGE values, we also have looked to the simulated recession shape and added the constraint of minimizing the differences (in terms of least squares) with the observed one.

We propose to change paragraph "3.2.4 Glacier parameterization" and replace it with the following paragraph:
p.10 l. 6-10 : "In this study, storage of liquid water inside glaciers was implemented by adding an englacial porous layer between the glacier and the bedrock, allowing the liquid water storage within the glacier. This englacial porous layer has a depth of 2 m and is characterized by a porosity of 0.8 and a hydraulic conductivity (vertical and lateral) of $3 \cdot 10^{-4}$ m/s (see Table A2). These parameters are kept constant through the simulations and were optimized according to the constraint of minimizing the differences (in terms of least squares) between the recession shape of the simulated hydrographs and the observed one."

The sentitivity to the englacial porous layer parameter values is also evaluated and analyzed in section 5.2.3 "Sensitivity to model parameters" p.30

- Sensitivity analysis to the parameter values

A sensitivity analysis to the parameters values was developed (5.2.3 "Sensitivity to model parameter values" p.30) in order to have a hint on the expected uncertainty on the results. As our simulations with DHSVM-GDM require high computation resources, we were unfortunately unable to perform a true Monte Carlos analysis with a large number of simulations.However, we compared the results of the control run (v3) with the results obtained from 28 additional simulations with changing parameter values. This section discusses the impact of the parameter values on the simulated daily discharge and the estimated relative contributions to the outflow.

- Avalanche module

A sentence was added (p.9 l.15-16) to explain how snow is ditributed in case of several possible downslope directions :
"In case of several possible directions downstream, avalanche snow is distributed according to a ratio based on the slope of each of the directions."

We hope that with these modifications you will be able to consider our paper for publication.

Best regards,

[revised manuscript text omitted]

---

## Author Response (AR4)

Editor report:

Dear authors,

The revised version of the paper is now close to be accepted. However, I would like to see the following additions to the new sections:

p.10 l. 6-10 : ....These parameters are kept constant through the simulations and were optimized according to the constraint of minimizing the differences (in terms of least squares) between the recession shape of the simulated hydrographs and the observed one." - It would be helpful to the reader to cite some references that discussed the assumption of keeping these parameters constant of changing it in time as has been done in other models simulating the water storage within the glacier.

Authors response:

Dear Editor,

Please find herewith a modified version of our paper "Quantification of different flow components in a high-altitude glacierized catchment (Dudh Koshi, Himalaya): some cryospheric-related issues".

References were added to the manuscript to discuss the assumption of using constant parameters for simulating the englacial storage. The modifications that were applied to the manuscript are the following:

p.10 l.4-12 (Sect. 3.2.4 Glacier parameterization): "Previous studies have shown the good performance of adding a conceptual representation of the storage and drainage in the glaciers within glacio-hydrological models (e.g., Jansson et al., 2003, Hock and Jansson, 2006). The most widely adopted approach is based on a reservoir or a cascade of reservoirs with time-invariant parameters (e.g., Farinotti et al., 2012; Zhang et al., 2015; Hanzer et al., 2016; Gao et al., 2017). Here, storage of liquid water inside glaciers was implemented by adding an englacial porous layer between the glacier and the bedrock allowing the liquid water storage within the glacier. This englacial porous layer has a depth of 2 m and is characterized by a porosity of 0.8 and a hydraulic conductivity (vertical and lateral) of $3 * 10^{-4}$ m/s (see Table A2). As in the previously cited studies, the parameters are kept constant through the simulations. They were optimized here according to the constraint of minimizing the differences (in terms of least squares) between the recession shape of the simulated hydrographs and the observed one."

p.28 l.16-23 (Sect. 5.2 Representation of the cryospheric processes in the model): "The results presented in this study also indicate potential future improvements to increase the reliability and to reduce the uncertainty of the simulations at short time steps. While at daily and longer time scales the different hydrological components seem to be well reproduced by the model, the analysis of the diurnal cycle (Fig. 15) shows that DHSVM-GDM responds too rapidly to the ice melt production with too high diurnal peak discharges. This is probably related to the use of constant parameters in the parameterization of the englacial porous layer for glacier storage. Taking into account the seasonal variation of the efficiency of the englacial drainage system appears necessary to simulate the diurnal flow cycle correctly (Hannah and Gurnell, 2001). Therefore, further improvements should be based on studies analyzing the mechanisms of glacier drainage systems in the Khumbu region and their influence on glacier outflows (e.g., Gulley et al., 2009; Benn et al., 2017). These studies …"

Best regards,